# Creation of Wildfire Susceptibility Maps in Plumas National Forest Using InSAR Coherence, Deep Learning, and Metaheuristic Optimization Approaches

Arip Syaripudin Nur [1], Yong Je Kim [2] and Chang-Wook Lee [1,3,]*

1   Division of Science Education, Kangwon National University, Chuncheon-si 24341, Korea
2   Department of Civil and Environmental Engineering, Lamar University, 4400 MLK Blvd., Beaumont, TX 77710, USA
3   Department of Smart Regional Innovation, Kangwon National University, Chuncheon-si 24341, Korea
*   Correspondence: cwlee@kangwon.ac.kr

**Abstract:** Plumas National Forest, located in the Butte and Plumas counties, has experienced devastating wildfires in recent years, resulting in substantial economic losses and threatening the safety of people. Mapping damaged areas and assessing wildfire susceptibility are necessary to prevent, mitigate, and manage wildfires. In this study, a wildfire susceptibility map was generated using a CNN and metaheuristic optimization algorithms (GWO and ICA) based on images of areas damaged by wildfires. The locations of damaged areas were identified using the damage proxy map (DPM) technique from Sentinel-1 synthetic aperture radar (SAR) data collected from 2016 to 2020. The DPMs' depicting areas damaged by wildfires were similar to fire perimeters obtained from the California Department of Forestry and Fire Protection (CAL FIRE). Data regarding damaged areas were divided into a training set (50%) for modeling and a testing set (50%) for assessing the accuracy of the models. Sixteen conditioning factors, categorized as topographical, meteorological, environmental, and anthropological factors, were selected to construct the models. The wildfire susceptibility models were evaluated using the area under the receiver operating characteristic (ROC) curve (AUC) and root mean square error (RMSE) analysis. The evaluation results revealed that the hybrid-based CNN-GWO model (AUC = 0.974, RMSE = 0.334) exhibited better performance than the CNN (AUC = 0.934, RMSE = 0.780) and CNN-ICA (AUC = 0.950, RMSE = 0.350) models. Therefore, we conclude that optimizing a CNN with metaheuristics considerably increased the accuracy and reliability of wildfire susceptibility mapping in the study area.

**Keywords:** wildfire; Plumas National Forest; damage proxy map (DPM); metaheuristic optimization; susceptibility map

## 1. Introduction

A wildfire is any uncontrolled fire that spreads through vegetative fuels and threatens to destroy life, property, or resources [1]. In California, wildfires are one of the most common natural disasters and have caused significant harm to the environment, society, and economy in recent years [2]. Particularly in the last two decades, climate change and land utilization for human activities have exacerbated wildfires [3]. In addition, sustained population growth and rapid social development have led to the expansion of areas of the wildland–urban interface (WUI) and have seriously increased the number of individuals and buildings affected by wildfires, which has further exacerbated their negative impact on human life [4]. The number of acres burned, number of fires, and amount of property damaged due to wildfires have not substantially decreased in California, despite significant administrative investment in wildfire suppression and management, according to the Wildfire Redbook published by the California Department of Forestry and Fire Protection (CAL FIRE).

According to wildfire activity statistics from CAL FIRE in 2020, wildfire agencies responded to an average of 8198 fires that burned 7263 km$^2$ per year from 2016 to 2020 [1]. Three of the deadliest, largest, and most destructive California wildfires occurred in Butte and Plumas County [5]. The Dixie Fire was caused by powerlines and burned 3898 km$^2$, killing one person and damaging 1329 structures in July 2021. The North Complex Fire was caused by lightning and burned 1291 km$^2$, took 15 lives, and destroyed 2352 structures in August 2020. Another devastating wildfire that occurred in November 2018 was the Camp Fire, which burned 621 km$^2$, claimed 85 lives, and destroyed 18,804 structures. The secondary effects of wildfires include health problems, reduced air quality, and economic losses due to wildfire evacuation [6–8]. In terms of long-term impacts, wildfires discharge aerosols and produce carbon emissions that affect the climate and climate change-related phenomena that drive even more wildfires [9].

Taking into account the various impacts caused by the burning and spreading of wildfires, collecting data on past and current wildfires, including their start date, location, amount of area burned, and duration, serves as valuable information for developing planning programs for the prevention and response to wildfires [10]. These inventories are an essential source of information for creating a hazard, risk, and susceptibility map. The identification of wildfire-prone areas can be achieved through susceptibility mapping [11–13]. Susceptibility is the likelihood of a wildfire in the future in the spatial dimension, with the predisposition to burn for each spatial unit (grid cell) assessed based on terrain, meteorological, and anthropogenic features such as topography and land cover, with the risk ranked from low to high [10]. This study evaluated past and recent wildfires in particularly damaged areas and determined the anthropogenic and geo-environmental predisposing factors that increase wildfire risk.

Providing accurate data regarding the location of previous wildfires is a critical initial step in susceptibility mapping. Wildfire inventory maps can be generated by several methods, including extensive global positioning system (GPS) field surveys [12,14] or remote sensing data [11,15]. Moderate-resolution imaging spectroradiometer (MODIS) products are freely accessible and are widely employed for generating datasets for wildfire-affected areas [11,14–16]. However, the spatial resolution of the hotspot used for identifying wildfires is 1 km. Sulova et al. (2020) combined a MODIS dataset and a Sentinel-2 mission to obtain the locations of wildfires with a spatial resolution of 20 m. Sentinel-2 is an optical satellite with 13 spectral bands that can be utilized for fuel condition mapping (e.g., vegetation greenness, fuel moisture) and burn severity mapping. However, its function depends on the atmospheric conditions of the areas studied. Radar satellite has several advantages over optical sensors: clouds and smoke are transparent to radar signals; radar does not require sunlight; and due to its coherent character, radar signal has the ability to detect minor alterations in surface property changes. The potential of synthetic aperture radar (SAR) images for mapping burnt areas lies in the sensitivity of SAR backscattering to vegetation structure and biomass, and the changes in scattering modes caused by fire events. The InSAR technique can make a direct measurement on the decorrelation among different acquisition dates by integrating both amplitude and phase information [17]. For these reasons, this study utilized the SAR dataset and the damage proxy map (DPM) method to identify areas of Plumas National Forest that have been likely damaged by wildfires and to generate an inventory dataset. The DPM method was developed by a joint project between the California Institute of Technology and the Jet Propulsion Laboratory (JPL), known as the Advanced Rapid Imaging and Analysis (ARIA) program. Using the DPM method is beneficial for damage mapping following natural disasters and has been applied to events including the 2017 Pohang earthquake in South Korea [18], Typhoon Hagibis in Japan [19], and the 2014 eruption of the Kelud volcano in Indonesia [20].

Various probabilistic and statistical approaches have been used to assess wildfire susceptibility in recent decades, including the frequency ratio (FR) [21,22], Bayesian theory [23,24], analytic hierarchy process (AHP) [25,26], and logistic regression (LR) [27,28]. Machine learning has also gained popularity for generating wildfire susceptibility maps, including artificial neural networks (ANNs) [29], support vector machines (SVMs) [30,31],

decision tree-based algorithms [12], and random forests [11,13]. These machine learning methods show great potential for evaluating non-linear and multivariate datasets. However, previous machine learning methods could not uncover more representative features from the input data to increase the precision of the forecasting process. Therefore, a new robust method has been developed to address this limitation by generating wildfire susceptibility mappings using deep learning algorithms, for instance, deep neural networks (DNNs) [32] and convolutional neural networks (CNNs) [33,34]. The CNN algorithm has a greater ability to recognize high variability patterns compared with other deep learning algorithms [35]. However, the choice of parameter settings has a substantial impact on the learning process, impacts the prediction results, and potentially leads to problems of overfitting or underfitting. Metaheuristic optimization based on iterative simulation is a widely used technique for addressing these problems; by optimizing the hyperparameters, it exhibits better performance than standalone machine learning and has been employed in the analysis of disasters such as landslides [36], subsidence [37], and floods [38].

This study aimed to produce wildfire susceptibility maps for Plumas National Forest by combining SAR data and deep learning methods based on CNN with metaheuristic optimization algorithms. Firstly, we generated DPMs after past wildfires occurring from 2016 to 2020, based on the Sentinel-1 SAR dataset as a dependent variable. The independent variables in the analysis were 16 conditioning factors from four categories: topographical, meteorological, anthropological, and environmental. Deep learning and metaheuristic optimization approaches were used in the analysis. Model performance was evaluated using root mean square error (RMSE) analysis and area under the receiver operating characteristic curve (AUC) analysis. Finally, high-risk and safe areas in Plumas National Forest were identified by creating maps based on the model.

## 2. Materials and Methods

Figure 1 shows the overall method that has been summarized in a graphical illustration. The first step is generating wildfire inventory utilizing the DPM and SAR datasets. The DPM-derived wildfire inventory data were then randomly separated into training (50%) and testing (50%) sets. After that, a spatial database was produced and then assessed employing spatial correlation analysis using the generated wildfire inventory and layers of the related factors.

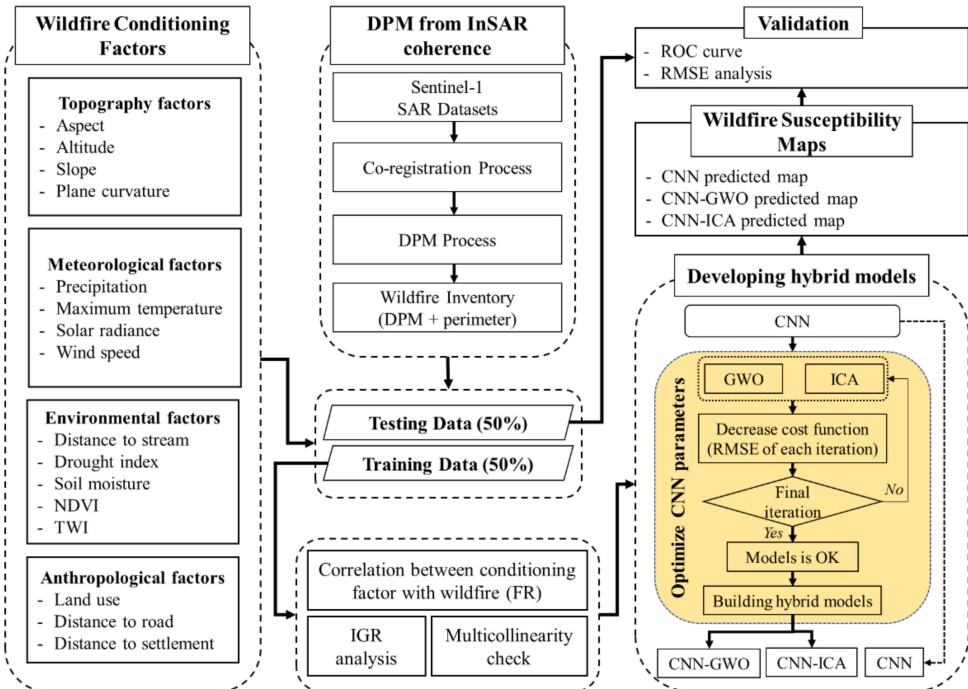

**Figure 1.** Flowchart of the overall methodology.

Next, the hyperparameters of CNN were optimized using GWO and ICA metaheuristic optimization algorithms and wildfire susceptibility models were produced. Finally, the wildfire susceptibility maps were compared and evaluated using RMSE and AUC analysis.

### 2.1. Study Area

Plumas National Forest resides in 4638 km$^2$ of mountain lands in the north of Sierra Nevada. Approximately 85% of the forest sits in Plumas County and the rest extends into Butte County, CA, USA, as shown in Figure 2. The forest was named after its main watershed, the Rio de las Plumas or Feather River. This area has a Mediterranean climate with hot, dry summers and cool, wet winters. Plumas National Forest has a complex topography and heterogeneous and dense vegetation (92.62% of its surface is covered by forests).

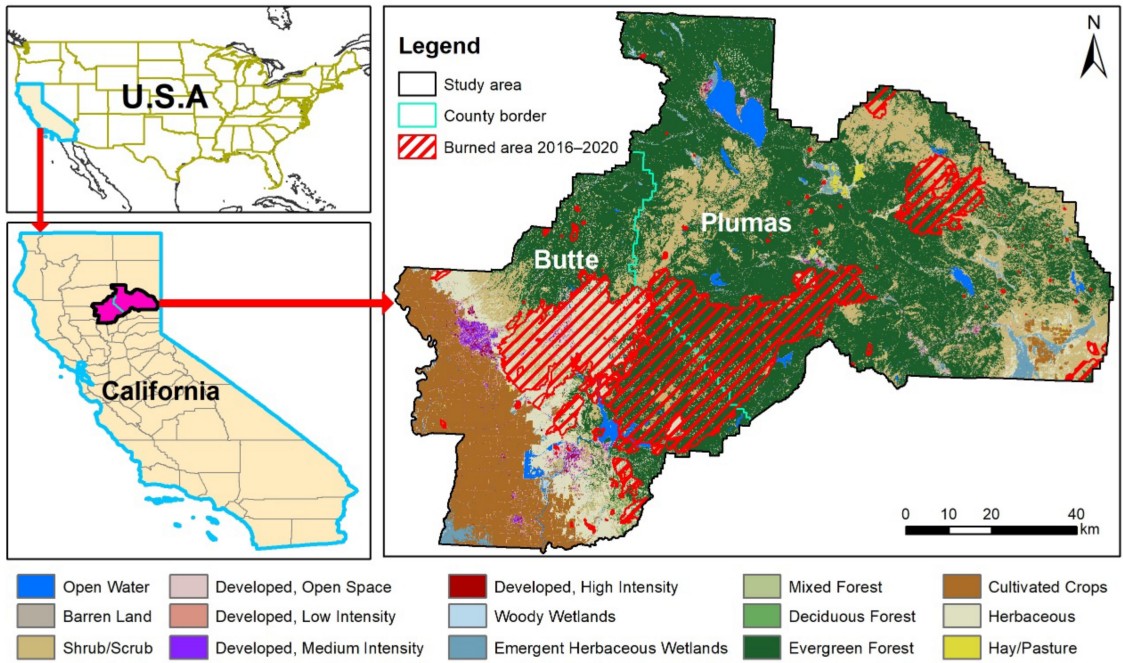

**Figure 2.** The land use of the study area located in California, United States of America, and the distribution of historical wildfires from 2016 to 2020 in Plumas National Forest.

The Fire and Resources Assessment Program (FRAP) created fire perimeters and established a database to represent a comprehensive digital record of fire perimeters in California [1]. Figure 2 shows the distribution of historic fires in Plumas National Forest from 2016 to 2020. There were 77 wildfires in Plumas National Forest from 2016 to 2021, with an average of 13 wildfires burning 115,711 ha per year. Figure 3 shows the number of wildfires and burned areas from 2016 to 2021. The greatest number of wildfires occurred in 2020, with 23 fires, while the largest total burned area was 433,100 ha in 2021. Although the number of wildfires in Plumas National Forest is uncertain, the burned area is widening every year. In general, the peak season for wildfires in Plumas National Forest is late summer and early autumn. In term of frequency, from 2016–2020, the wildfire season started in April and the most frequent occurrence was observed in August.

CAL FIRE [1] also investigated and recorded the causes of wildfires and found that human causes (direct or indirect) ignited 38.57% of the wildfires in Plumas National Forest (as shown in the statistical summary in Table 1). Therefore, anthropological factors should be considered in constructing a wildfire susceptibility map. According to the report, the unknown ignition cause describes a fire that has been investigated and has insufficient information to classify further, a fire that is under investigation, or a fire that has not yet been investigated.

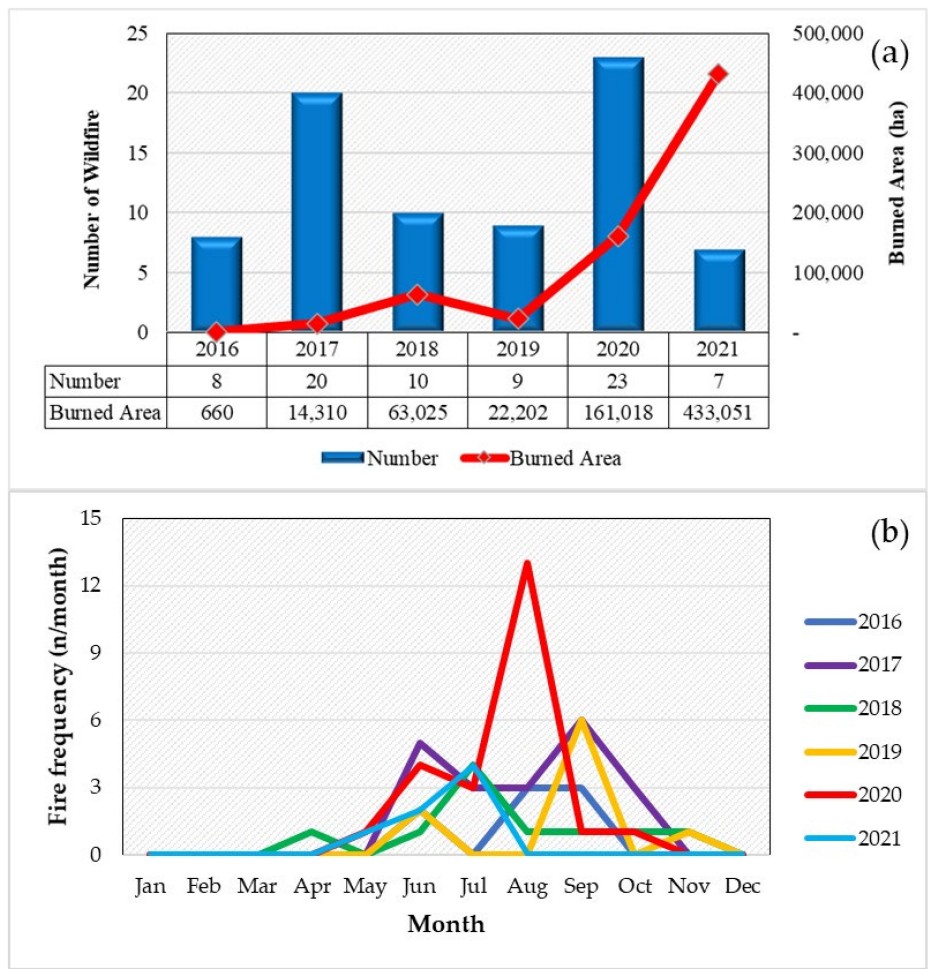

**Figure 3.** (**a**) Yearly frequency of wildfires and burned areas and (**b**) fire frequency by month in Plumas National Forest from 2016 to 2021.

**Table 1.** Statistical summary of wildfire ignition causes in Plumas National Forest from 2016 to 2020. (2021 wildfires are under investigation).

| Causes | Lightning | Human-Caused | | | Miscellaneous | Unknown |
|---|---|---|---|---|---|---|
| | | Transportation | Human Activity | Construction | | |
| Number of wildfires | 25 | 3 | 18 | 6 | 8 | 10 |
| Percentage | 35.71 | 4.29 | 25.71 | 8.57 | 11.43 | 14.29 |

### 2.2. SAR Datasets

SAR is beneficial compared to other types of remote sensing because its radar transmission in the microwave spectrum is not affected by day and night visibility and weather conditions, including cloud cover or smoke from fires [17]. Sentinel-1 SAR C-band data (5.5 cm wavelength) from the European Space Agency (ESA) were utilized to depict damaged areas after a wildfire in Plumas National Forest. Plumas National Forest is located on path 42 and frame 462. We collected Sentinel-1 single look complex (SLC) data with interferometric wide-swath (IW) mode and vertical transmission and vertical return (VV) polarization. We collected three scenes of Sentinel-1 SLCs for each wildfire of more than 300 ha acres. Two images were obtained before the wildfire and one after the wildfire was contained. In total, we collected 33 scenes to depict the damaged areas after wildfires from 2016 to 2020 in Plumas National Forest (Table 2). The Sentinel-1 SLC level-1 data were downloaded from the Alaska Satellite Facility (ASF) where each scene was pre-processed in Gamma

2021 software, (https://www.gamma-rs.ch/uploads/media/upgrades_info_20211201.pdf (accessed on 12 July 2022)). Each scene was updated through orbit information update to reduce orbit error.

**Table 2.** Wildfire information and Sentinel-1 SAR data used in this study.

| Fire Name | Alarm Date (MM/DD/YYYY) | Burned Area (ha) | Pre-Event | | Post-Event | Flight Direction |
|---|---|---|---|---|---|---|
| | | | 1 | 2 | 3 | |
| North Complex | 08/17/2020 | 129,004 | 08/01/2020 | 08/13/2020 | 09/06/2020 | Desc |
| Sheep | 08/17/2020 | 11,967 | 08/01/2020 | 08/13/2020 | 08/25/2020 | Desc |
| Loyalton | 08/14/2020 | 19,032 | 08/01/2020 | 08/13/2020 | 09/06/2020 | Desc |
| Walker | 09/04/2019 | 22,107 | 08/20/2019 | 09/01/2019 | 09/13/2019 | Asc |
| Camp | 11/08/2018 | 62,053 | 10/23/2018 | 11/04/2018 | 11/16/2018 | Desc |
| Cascade | 10/08/2017 | 4042 | 09/10/2017 | 10/04/2017 | 10/16/2017 | Desc |
| Cherokee | 10/08/2017 | 3406 | 09/10/2017 | 10/04/2017 | 10/16/2017 | Desc |
| Ponderosa | 08/29/2017 | 1625 | 08/06/2017 | 08/18/2017 | 09/11/2017 | Asc |
| Minerva 5 | 07/29/2017 | 1744 | 07/01/2017 | 07/13/2017 | 08/06/2017 | Asc |
| Wall | 07/07/2017 | 2441 | 06/18/2017 | 06/30/2017 | 07/12/2017 | Desc |
| Saddle | 09/05/2016 | 344 | 080/5/2016 | 08/29/2016 | 09/22/2016 | Asc |

## 2.3. Wildfire Conditioning Factors

Selecting independent variables, also known as predictors, predisposing, or conditioning factors, is vital in predictive modeling. Based on prior studies in California [2,39,40], 16 factors related to wildfire susceptibility were selected and categorized into four categories as shown in Table 3: topographical, meteorological, environmental, and anthropological. The topographical factors included aspect, altitude, slope, and plan curvature; the meteorological factors included precipitation, maximum temperature, solar radiance, and windspeed; the environmental factors included distance to stream, drought index, soil moisture, NDVI, and topographic wetness index; and the anthropological factors included land use, distance to road, and distance to settlement. The 16 factors used for the wildfire susceptibility analysis are shown in Figure 4. All factors were organized into a raster-based spatial database with a spatial resolution of 30 m. Continuous (numeric) data were reclassified into five classes using the quantile method to identify and analyze the effect of wildfires.

**Table 3.** The list and description of wildfire conditioning factors.

| Category | Factors | Scale/Resolution | Source of Data | References |
|---|---|---|---|---|
| Topography | Aspect<br>Altitude<br>Slope<br>Plan curvature | 30 m | Copernicus DEM | [2]<br>[40]<br>[16]<br>[14] |
| Meteorological | Precipitation<br>Maximum temperature<br>Solar radiance<br>Windspeed | 800 m<br><br>4 km<br>100 m | PRISM<br><br>NREL<br>Global Wind Atlas | [2]<br>[39]<br>[41]<br>[42] |
| Environmental | Distance to stream<br>Drought index<br>Soil moisture<br>NDVI<br>Topographic wetness index | 1:5000<br><br>4 km<br><br>375 m<br>30 m | California State Geoportal<br><br>Terra Climate<br><br>MODIS<br>Copernicus DEM | [16]<br>[43]<br>[44]<br>[39]<br>[40] |
| Anthropological | Land use<br>Distance to road<br>Distance to settlement | 30 m | USGS | [2]<br>[14]<br>[41] |

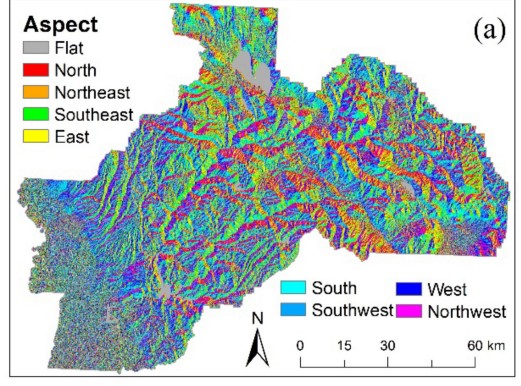

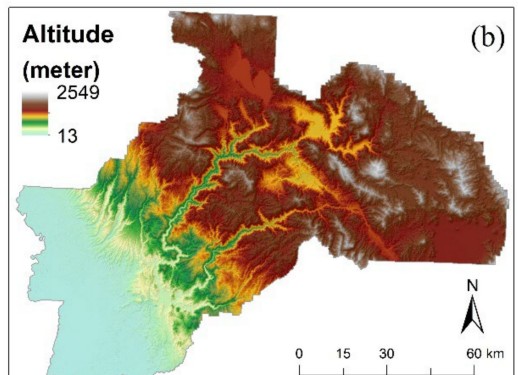

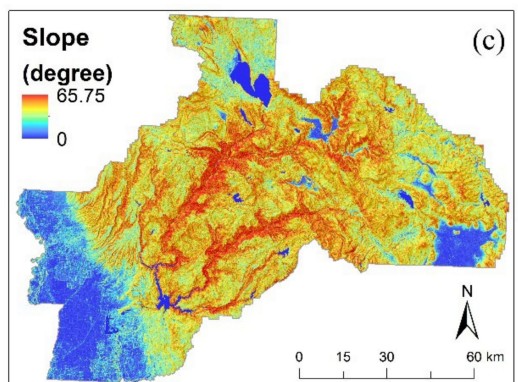

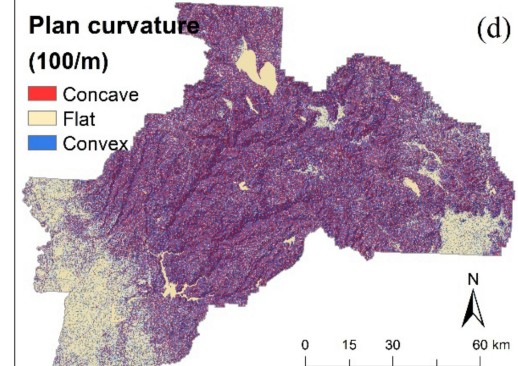

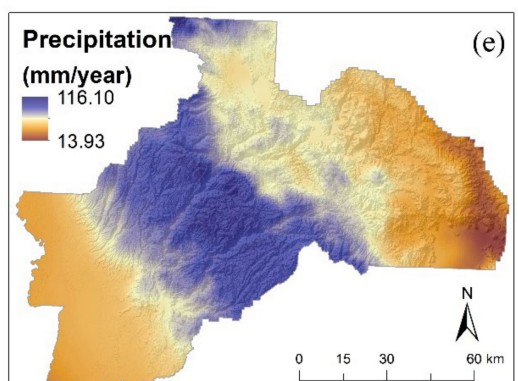

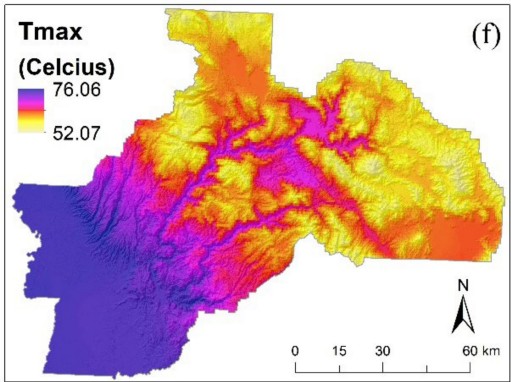

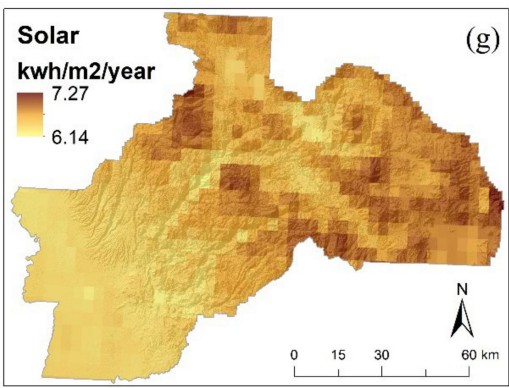

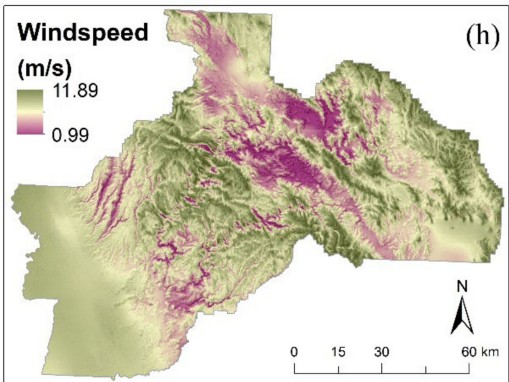

**Figure 4.** *Cont.*

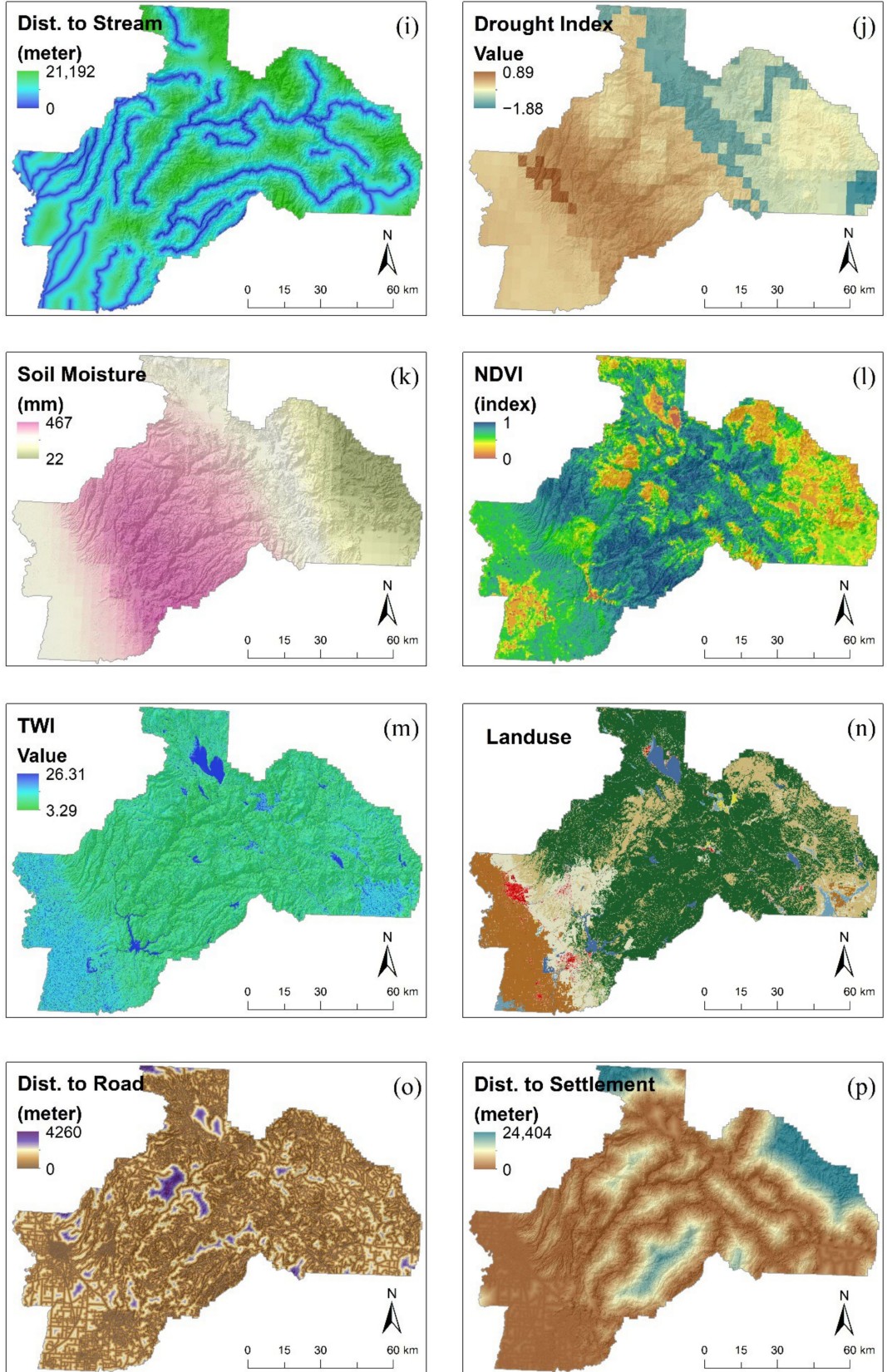

**Figure 4.** Wildfire conditioning factors used in this study categorized as (**a**–**d**) topographical, (**e**–**h**) meteorological, (**i**–**m**) environmental, and (**n**–**p**) anthropological factors.

The topographic conditioning factors were altitude, aspect, slope, and plan curvature, derived from the Copernicus digital elevation model (DEM) with a spatial resolution of 30 m. The Copernicus DEM is a product of radar satellite data collected during the TanDEM-X mission. Altitude is an important factor affecting the spread and severity of a wildfire and is associated with local climate variations, vegetation distribution, composition, and flammability [14,45]. Increasing the degree of inclination can increase the rate of fire spread. Fire can spread quickly into steep areas and less quickly down steep areas [11]. Aspect shows the direction the slope is facing and affects the amount of solar radiation received [15,40]

The meteorological conditioning factors were the mean values of precipitation, maximum temperature, solar radiation, and wind speed. These factors control the life cycle of flora and fauna, which contribute to producing fuels, drying fuels for ignition, or spreading wildfires [15]. Solar radiation was collected from the National Renewable Energy Laboratory (NREL) with a spatial resolution of 4 km [46]. Increasing solar radiation can reduce soil moisture and increase temperature, thus increasing wildfire risk [47]. Precipitation and maximum temperature were collected from parameter-elevation regressions on independent slopes model (PRISM) managed by Oregon State University's PRISM Climate group. Precipitation affects the moisture level and vegetation pattern that influence the speed of fire distribution [48]. There is a direct correlation between increases in temperature and wildfires [14,41]. Windspeed was obtained from the Global Wind Atlas with a spatial resolution of 100 m. Windspeed is related to wildfires, as wind affects their speed and severity [49]. The rasters were generated from images collected from 2016 to 2020 using the mean statistic function [43].

The environmental factors were the distance to stream, soil moisture, drought index, NDVI, and topographic wetness index (TWI). Stream data were acquired from California State Geoportal. The distance to stream has a direct role in forest health, with the stream serving as a water resource [14,50]. The soil moisture and drought index were collected from the Terra Climate 2016–2020 dataset with a spatial resolution of 4 km. This monthly dataset for global terrestrial surfaces uses the interpolated time-varying anomalies from CRU Ts 4.0/JRA55, with high-spatial climatological normal obtained from the WorldClim dataset to create a high-spatial resolution dataset that covers a broader temporal record [51]. The drought index, based on the Palmer Drought Severity Index (PDSI), measures agricultural drought calculated by adding precipitation to the top two layers of the soil and employing a temperature-driven evapotranspiration algorithm to remove moisture [13]. Soil moisture directly influences the dryness of fuels and affects the dead fuels generally found in the ground; thus, it acts as a proxy for drought [52]. The normalized difference vegetation index (NDVI) was collected from MODIS/Terra with a spatial resolution of 375 m. Instead of using the mean NDVI, we calculated the median NDVI from 2016 to 2020 on a pixel-by-pixel basis to avoid bias introduced to the mean value by the greenness loss after wildfire [53]. NDVI displays the health status of vegetation. A reduction of the NDVI indicates dry grass or trees affecting the water stress and increasing the risk of fire [54]. The topographic wetness index (TWI) was generated from the Copernicus DEM and calculated with the relevant equation from Hong et al. [55]. TWI defines the aspect of steady-state soil wetness [56].

The anthropological factors were land use, distance to road, and distance to settlement. Land use and distance to settlement were acquired from the United States Geological Survey (USGS) national land cover database (NLCD-2019) with a spatial resolution of 30 m. Land use describes the landscape features of the study area. The different characteristics, such as the load and moisture content of the distinct land use types, can influence the ignition and spread of fires [11]. The distance to settlements and roads quantifies access to forest areas and, in many cases, plays an essential role in the ignition of wildfire incidents [15,41,50].

### 2.4. Damage Proxy Map (DPM)

To help identify damaged areas after wildfires, DPMs were produced by comparing pre- and co-event SAR images. The method depends on the decrease in the coherence of the signal scattering between SAR images taken prior and following the event to determine irregular changes in ground surface properties [57]. Coherence assesses the alteration in backscatter signal as a proxy for the ground-surface property alteration [58]. A high coherence implies little or no alteration to the ground surface reflecting the SAR radiation. Alterations can be caused by damage to the ground surface itself or damage to structures by wildfires.

Interferometric coherence is a quantification of resemblance between two radar signals that have been employed to assess the quality of the InSAR product. This statistical quantity is computed as [19]:

$$\gamma = \frac{|\langle c_1 c_2^* \rangle|}{\sqrt{\langle c_1 c_1^* \rangle \langle c_1 c_2^* \rangle}}, 0 \leq \gamma \leq 1 \tag{1}$$

where $c_1$ and $c_2$ represent the values of complex pixels of two SAR scenes and * indicates the complex conjugate. The resulting coherence ranges from 1 (coherent) to 0 (incoherent). If the two images' observations are identical, then the coherence is equal to 1 due to stable objects in the scenes, such as buildings. However, the DPM method is not limited to building damage. As long as the predamage (reference) InSAR pair maintains reasonable coherence, this approach can be applied to any anthropogenic or natural damage that causes serious surface property changes, such as those caused by a wildfire. Major damage after a wildfire significantly intensifies the interferometric phase dissimilarity. This alteration appears as a decrease in coherence or decorrelation.

The process started with image co-registration with subpixel accuracy utilized to match scenes to one another. Three radar images were then employed to generate two pairs of interferometric coherence maps to produce a DPM: one pair prior to the damage (pre-event) and the other pair spanning the damage (co-event). The pre-event coherence pair delineated changes unrelated to the event and was expected to be the background value. In ArcGIS 10.4 software, we obtained the coherence difference (COD) by subtracting $\gamma_{\text{pre-event}}$ and $\gamma_{\text{co-event}}$. The results were transformed into a georeferenced coordinate system employing the Copernicus DEM, resulting in pixels approximately 30 m across. The threshold for sufficient coherence loss was adjusted by comparing observed COD with reported damage and the area where there was no damage [58–60]. Here, the coherence loss threshold for DPM was chosen by considering wildfire perimeters from CAL FIRE. The fire perimeter was obtained using various collection methods, such as GPS ground, GPS air, infrared, photo interpretation, hand-drawn, and mixed collection methods.

### 2.5. Spatial Correlation Analysis

Variable selection is particularly important in the prediction of wildfire susceptibility. The high dimensionality of the training dataset may complicate the prediction process and decrease the prediction accuracy. In this study, the information gain ratio (IGR) and multicollinearity analysis were selected to evaluate the wildfire conditioning factors.

In addition, Pearson correlation analysis was applied to identify linear correlation relationships between pairs of variables. The Pearson correlation coefficient allows measurement of the association between variables. When a correlation is present, a change in the magnitude of one variable is associated with a change in the magnitude of the other variable, either in the same direction (positive coefficient) or in the opposite direction (negative coefficient). This coefficient is scaled and takes values between −1 and 1, where 0 is equivalent to the case in which no correlation exists [61]. A correlation coefficient greater than or equal to 0.7 is considered a correlation indicator that can lead to distortion of the modeling process and affect future predictions [62].

For the IGR method, the factor with a higher IGR value indicates a stronger predictive ability of the model. However, factors with IGR values equal to or less than 0 indicate

a "null" contribution to the forest fire susceptibility model and should be excluded from further analysis [34].

The tolerance (TOLs) and variance inflation factors (VIFs) were computed to discover whether multicollinearity existed [63]. Multicollinearity occurs between variables if they have a close linear association in a regression model, which may decrease the model's performance [64,65]. The TOL and VIF were calculated using the following equations:

$$TOL = 1 - R_j^2 \tag{2}$$

$$VIF = \frac{1}{1 - R_j^2} \tag{3}$$

where $R_j^2$ denotes the regression value of $j$ on different variables in a dataset. Thus, multicollinearity issues generally occur if the *TOL* score is smaller than 0.10 and the *VIF* score is greater than 10 [66,67].

Using the frequency ratio (FR) method, spatial relationships among the distribution of wildfire damage locations (71,371 points) and class of wildfire conditioning factors were analyzed. After classifying the numeric factors into five classes, tabulate area tools in ArcGIS 10.4 were used to handle dependent data (wildfire location) with classified factors for obtaining tabulate area cell counts indicating the number of wildfire occurrences in each class. Therefore, the FR value can be acquired by calculating the ratio between the percentage of pixels wildfire occurrence/event in each class with the percentage of pixels area of each class. If a specific class of the factor obtained a FR value greater than 1, then the class has a high probability of wildfire occurrence, high degree of damage by wildfire, and will have a substantial impact on wildfire susceptibility [68]. The FR value of each class factor was computed using the following equation:

$$FR = \frac{\%\ Pixel\ of\ wildfire\ occurrence}{\%\ Pixel\ class\ of\ wildfire\ factors} \tag{4}$$

The *FR* value of each class of wildfire conditioning factors was utilized to produce wildfire susceptibility maps using deep learning algorithms based on CNN, CNN-GWO, and CNN-ICA.

### 2.6. Convolutional Neural Network (CNN)

Classification and prediction using CNNs have been increasingly applied recently in numerous disciplines as well as earth sciences [69–71]. CNN is one of the deep learning approaches differentiated from a traditional neural network by using multiple layers such as pooling, mutual weights, and local connections [72]. Generally, the CNN architecture is constructed of convolutional layers (CLs), activation process (AP), pooling layers (PLs), and fully connected layers (FCLs). The CLs continuously identify the connection between each feature class by extracting new features from the input images [73]. Next is AP, which chooses a rectified linear unit due to the unit having the ability to handle non-linear results from CLs and enhance the non-linear properties of the neural network [35,74]. PLs allow for stable conversion to prevent overfitting and upgrade computational performance by decreasing the number of feature class structures through the down-sampling approach [75]. Last are FCLs, which combine data in the neural network and generate the final output [73,76].

### 2.7. Metaheuristic Optimization Algorithms
#### 2.7.1. Grey Wolf Optimization (GWO)

A metaheuristic algorithm was developed by mimicking the hierarchy of command and hunting strategy of grey wolves (Canis lupus) in nature [77]. The hierarchical structure of the grey wolf pack is based on four levels: alpha, beta, delta, and omega. Alpha ($\alpha$) is the leader that controls all activities of the pack, such as resting, migration, feeding, and hunting. Beta ($\beta$) assists the leader in making judgements and establishing the order of

the pack. Deltas ($\delta$) act as hunters, sentinels, and watchers. Omegas ($\omega$) are the weakest relations and care for the young; internal fights and other issues in the pack are noticed without them [78]. There are three main steps in hunting: seeking prey, surrounding the prey, and striking the prey [36]. The alpha ($\alpha$) is considered the highest-fitted solution, followed by $\beta$, $\delta$, and $\omega$, based on Equations (5)–(8) [77]:

$$\vec{D_\alpha} = |\vec{C_1} \cdot \vec{X_\alpha} - \vec{X}|, \ \vec{D_\beta} = |\vec{C_2} \cdot \vec{X_\beta} - \vec{X}|, \ \vec{D_\delta} = |\vec{C_3} \cdot \vec{X_\delta} - \vec{X}|, \ \vec{C} = 2 \times \vec{r_2} \tag{5}$$

$$\vec{X_1} = \vec{X_\alpha} - \vec{A_1} \times (\vec{D_\alpha}), \ \vec{X_2} = \vec{X_\beta} - \vec{A_2} \times (\vec{D_\beta}), \ \vec{X_3} = \vec{X_\delta} - \vec{A_3} \times (\vec{D_\delta}), \tag{6}$$

$$\vec{A} = 2 \times \vec{a} \times \vec{r_1} - \vec{a} \tag{7}$$

$$\vec{X}(t+1) = \frac{\vec{X_1} + \vec{X_2} + \vec{X_3}}{3} \tag{8}$$

where for the iteration time t, $\vec{D}$ is the vector that suggests a new position and $\vec{X}$ denotes the position of the wolf; $\vec{A}$ and $\vec{C}$ are coefficient vectors and components of $\vec{a}$, which are linearly diminished between 0 and 2 in each iteration; and $\vec{r_1}$ and $\vec{r_2}$ are random vectors generated for the range [0, 1]. The hunting process is completed when $\vec{A}$ takes values between $-1$ and 1, when an attack occurs [78].

### 2.7.2. Imperialist Competitive Algorithms (ICA)

ICA is one of the metaheuristic optimization algorithms imitating imperialistic competition and addresses various optimization problems [79]. ICA begins with a population of randomized solutions, where each solution is considered as a country. The population is separated into two categories: imperialist and colonies. Each imperialist initially controls a group of colonies and then competes to attract and take possession of others through its authority and influence, forming the core of the ICA algorithm [80]. The positions of imperialist countries and colonies can be exchanged if the colonies become more powerful. If an empire cannot increase its strength through interactions with other countries, it will be gradually eliminated during the competition. Thus, fragile empires lose their colonies, while substantial empires occupy more colonies, further increasing their strength. Ultimately, imperialistic competition leads to the elimination of all but one empire, which gains control of all other countries [81]. The algorithm terminates after the maximum number of iterations or when a predefined number of empires remain.

### 2.8. Accuracy Assessment

Evaluation is an essential step in assessing the accuracy of predictions of a model to support the scientific validity of a study [82]. This study used the area under the receiver operating characteristic (ROC) curve analysis for model assessment, which is a common way of evaluating wildfire probability models [11,15,59,83,84]. The AUC represents the performance, evaluation, utilization, and compression of model predictions [85–87]. This measure of accuracy ranges between 0.5 and 1 (perfect forecasting), with values near 1 indicating excellent performance and values near 0.5 denoting very poor prediction accuracy [69]. We applied this method using the testing dataset from wildfire inventory pixels (see Section 3.1) which were not used to train the applied deep learning approaches. A ROC curve is a plot of specificity (i.e., false positive on the x axis) versus sensitivity (i.e., true positive on the y axis). In wildfire modeling, sensitivity and specificity refer to the proportion of correctly predicted fire pixels and the proportion of correctly predicted non-fire pixels, respectively.

This study also used the mean square error (MSE) and root mean squared error (RMSE) approach as a second evaluation metric and as a cost function for optimizing CNN parameters as follow:

$$MSE = \frac{1}{n} \sum_{i=1}^{n} (y_i - t_i)^2 \tag{9}$$

$$RMSE = \sqrt{\frac{1}{n} \sum_{i=1}^{n} (y_i - t_i)^2}$$

where $y$ and $t$ are the predicted and actual wildfire inventory values, respectively, and $n$ denotes the number of samples. Metaheuristic algorithms were used to minimize the *MSE* and *RMSE* values through optimization of model hyperparameters. Smaller values of *MSE* and *RMSE* indicate a better performance of the models and the effectiveness of metaheuristic optimization. In addition, a wildfire susceptibility map with higher performance will be compared with recent wildfires in 2021 to evaluate the model's predictive performance.

## 3. Results

### 3.1. Wildfire Damage Inventory Map

Most wildfires in Plumas National Forest occur during the summer or fall when most of the leaves have fallen. Thus, InSAR coherence measurements collected before a wildfire can maintain reasonable coherence, and the DPM technique can be applied to depict damaged locations after wildfires in Plumas National Forest.

DPMs were generated depicting areas in Plumas National Forest that were likely damaged by wildfires. Figure 5 reveals the DPM results after wildfires in Plumas National Forest, including the Camp Fire, Walker Fire, North Complex Fire, and Sheep Fire. Areas that experience an increase in COD are indicated by pixels with colors that stretch from yellow to red. Enlarged DPM pixels opaqueness represents significant ground or building alteration or possible damage due to wildfire. Regions where decorrelation did not alter significantly over wildfires during the period are set to be transparent, indicating no destruction. The DPMs were geocoded to the Copernicus DEM with a corresponding spatial resolution of approximately 30 m.

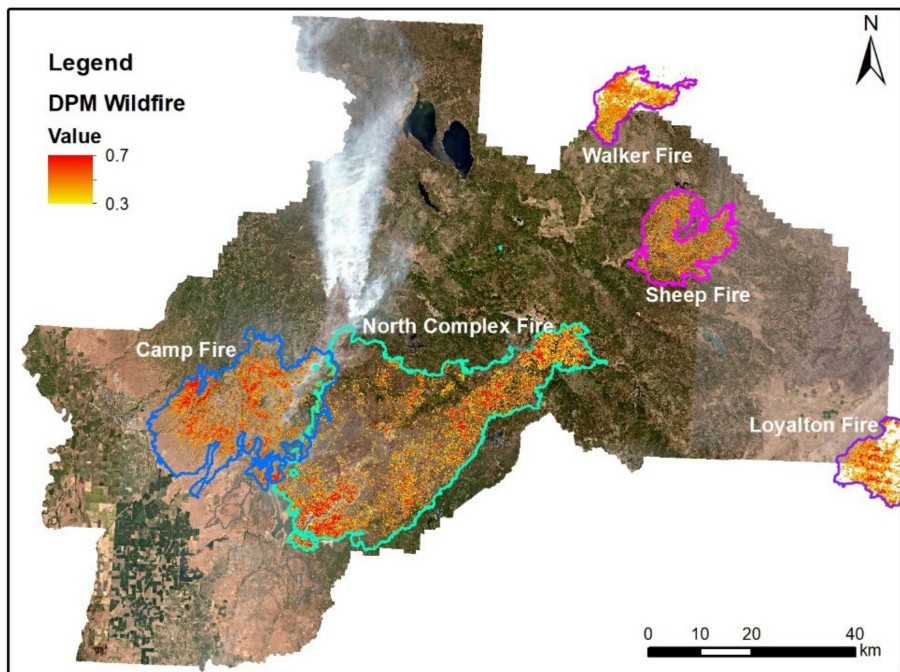

**Figure 5.** DPM depicting damage area after Camp Fire, Walker Fire, North Complex Fire, and Sheep Fire wildfires. Image from Sentinel-2 data acquired on 17 July 2021, during the Dixie Fire.

This study compared the distribution of coherence loss areas with the fire perimeter from CAL FIRE which showed similar damage areas after wildfires throughout Plumas National Forest from 2016 to 2020. The scattered pixels were removed using a fire perimeter to generate precise wildfire-affected areas. The combination of both sources resulted in a more accurate and dependable inventory dataset depicting the location of damage after wildfires. The total number of wildfires mapped was 11, which translated to 71,371 pixels of 30 m wildfire cell size as wildfire occurrence data. These data were then randomly divided into a 50% training and 50% testing dataset. In the context of deep learning, the creation of a wildfire susceptibility map required data of areas without wildfires. In this study, the same number of non-wildfire location data (71,371 points) were sampled through random selection by comparing the area with no prior wildfire and the very low probability class identified with the frequency ratio method. The analysis of zero data based on these results was an efficient way to aid the interpretation of the area [35]. The non-wildfire data were also divided into a training (50%) and testing (50%) dataset. The 50% training datasets for wildfire-occurrence and non-wildfire-occurrence location were then merged to generate wildfire susceptibility maps, with the remaining 50% of testing data from both datasets merged for evaluation of the performance [59,84,85].

### 3.2. Relationship between Damage Area and Related Factors

Figure 6 shows the results from the Pearson correlation analysis for all the conditioning factors. According to the pairwise correlation matrix, none of the correlation values among each conditioning factor were within the safe threshold. The highest correlation value was computed between solar and maximum temperature (0.63). The results suggested that there is no need to eliminate any conditioning factors and all factors would not cause distortion in the modeling process.

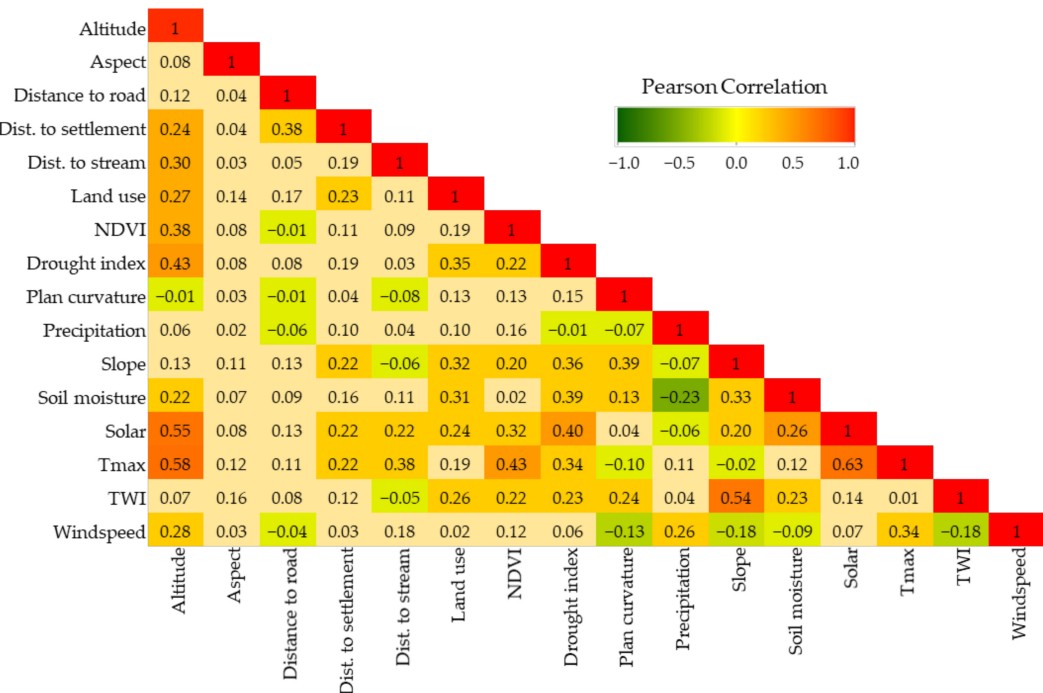

**Figure 6.** Pearson correlation between wildfire conditioning factors.

Based on the evaluation of the IGR analysis, all 16 conditioning factors considered in this study had predictive capabilities for wildfire modeling and contributed to the occurrence and extent of wildfires (IGR > 0), as shown in Table 4. Table 4 shows that the factor with the highest IGR value was land use (IGR = 0.39); thus, land use was the most effective wildfire conditioning factor in this study, followed by drought index (0.25) and

maximum temperature (0.18). Furthermore, windspeed was found to be the least important wildfire conditioning factor with an IGR value of 0.03.

**Table 4.** IGR and multicollinearity results for the conditioning factors.

| Factor | IGR | Collinearity Statistics | |
| --- | --- | --- | --- |
| | | TOL | VIF |
| Altitude | 0.17 | 0.17 | 5.91 |
| Aspect | 0.06 | 0.98 | 1.02 |
| Distance to stream | 0.06 | 0.99 | 1.01 |
| Distance to road | 0.14 | 0.97 | 1.03 |
| Distance to settlement | 0.03 | 0.68 | 1.47 |
| Land use | 0.39 | 0.39 | 2.55 |
| NDVI | 0.08 | 0.71 | 1.40 |
| Drought index | 0.25 | 0.24 | 4.13 |
| Plan curvature | 0.04 | 0.83 | 1.20 |
| Precipitation | 0.04 | 0.39 | 2.57 |
| Slope | 0.16 | 0.30 | 3.31 |
| Soil moisture | 0.18 | 0.27 | 3.66 |
| Solar | 0.12 | 0.41 | 2.49 |
| Maximum temperature | 0.18 | 0.22 | 4.63 |
| TWI | 0.16 | 0.53 | 1.90 |
| Windspeed | 0.03 | 0.81 | 1.22 |

An assessment of multicollinearity was conducted to investigate the correlation between wildfire conditioning factors, as shown in Table 4. All wildfire conditioning factors had VIF scores lower than 10 and tolerance scores >0.1. The ranges of VIF and TOL were within the permissible threshold; therefore, there were no multicollinearities observed between the wildfire conditioning factors, thereby avoiding the risk of deteriorating model performance. The maximum VIF score was 5.91 for altitude and the lowest tolerance score was 0.18 for maximum temperature. The range of TOL was 0.17 to 0.99. Given that the Pearson correlation analysis, IGR, and multicollinearity values were all within their critical value, all of the conditioning factors can be used for creating a wildfire susceptibility map.

FR values can provide information about the connection between the location of damage caused by wildfires and classes of conditioning factors. The results of the FR calculation in this study are shown in Table 5. The slope aspect showed that wildfire occurrence is concentrated in the northeast, southeast, south, southwest, west, and flat portions of Plumas National Forest. The fact that south-facing aspects receive more solar radiation, which increases fuel temperature, and low fuel moisture in the North Hemisphere led to wildfire occurrence [50,88]. The areas of 13–272 and 272–1216 m in altitude had high FR values of 1.03 and 1.54, respectively, indicating that low-altitude areas in the study area are more susceptible to wildfire occurrence and implying vegetation is more burnable due to high temperature and dry weather during summer [10,50]. The high degree of slope causes wildfires to spread more quickly up the steep areas, as shown by the two classes of 7.22–12.89 and 20.37–65.75 having FR values of 1.06 and 1.44, respectively. Higher distance to road indicates a higher probability of wildfire occurrence, with the three highest classes of 152–301, 301–595, and 595–4260 m having FR values of 1.19, 1.14, and 1.27, respectively. The further away the road is, the more difficult it is for firefighters to access and extinguish the wildfire area [15]. Portions of landscape with maximum temperature > 65.67 °C, a higher drought index of −0.09, and precipitation between 25–43.35 mm were predestined places for wildfires, encompassing 52.88% (Tmax > 65.57 °C), 53.20% (drought > −0.09–0.89), and 47.44% (25–43.35 mm precipitation). High NDVI classes, namely 0.46–0.61, 0.61–0.76, and 0.76–1, had FR values of 1.04, 1.10, and 1.14, respectively, implying wildfire occurrence in high greenness and the moisture content of vegetation. The reverse trends were found for distance to settlement, with the low classes showing high FR values of 1.28 and 1.30 for distance to settlement of 287–1148 and 1148–2679 m, respectively. A closer distance to

stream, namely 0–997, 997–2160, and 2160–3573 m, had high FR values of 1.17, 1.19, and 1.05, respectively. Streams cater for entertaining human interests such as tourist camps. The closer the distance to settlements and streams relating to high levels of human activity, the greater the risk of wildfire ignition [2,14]. For land use factors, herbaceous, evergreen, and mixed forest classes, with the highest FR values, experienced the highest wildfire occurrence and served as a fuel for wildfires [10,14]. Based on their FR values, plan curvature with concave and convex were associated with a high occurrence of wildfires, which is in agreement with other findings that suggest the probability of fire occurrence may be lower on flat terrain and higher on concave slopes [24]. Topographical and anthropological features also swamped the effect of soil moisture, TWI, and windspeed, all of which did not show any pattern or specific class that is strongly associated with the presence or absence of wildfires. The wildfire occurrence for TWI and windspeed features tended to be high in most classes, although our windspeed variable was intended to reflect the outcome of providing fresh oxygen and a greater potential for drying potential fuels and propelling fire across land at a faster rate.

**Table 5.** Frequency ratio of wildfire conditioning factors.

| Factor | Class | Total % | Event % | Frequency Ratio |
|---|---|---|---|---|
| Aspect | North | 1.85 | 0.00 | 0.00 |
| | Northeast | 10.30 | 10.58 | 1.03 |
| | East | 9.74 | 5.77 | 0.59 |
| | Southeast | 11.31 | 12.82 | 1.13 |
| | South | 12.56 | 13.78 | 1.10 |
| | Southwest | 14.31 | 16.03 | 1.12 |
| | West | 14.70 | 15.38 | 1.05 |
| | Northwest | 13.99 | 13.14 | 0.94 |
| | Flat | 11.25 | 12.50 | 1.11 |
| Altitude (m) | 13–272 | 19.83 | 20.51 | 1.03 |
| | 272–1216 | 20.13 | 31.09 | 1.54 |
| | 1216–1525 | 19.43 | 16.99 | 0.87 |
| | 1525–1773 | 20.46 | 16.99 | 0.83 |
| | 1773–2549 | 20.15 | 14.42 | 0.72 |
| Distance to Road (m) | 0–60 | 21.26 | 16.35 | 0.77 |
| | 60–152 | 20.13 | 13.46 | 0.67 |
| | 152–301 | 19.70 | 23.40 | 1.19 |
| | 301–595 | 19.47 | 22.12 | 1.14 |
| | 595–4260 | 19.44 | 24.68 | 1.27 |
| Distance to Settlement (m) | 0–287 | 17.90 | 12.82 | 0.72 |
| | 287–1148 | 22.34 | 28.53 | 1.28 |
| | 1148–2679 | 20.25 | 26.28 | 1.30 |
| | 2679–5646 | 20.08 | 16.67 | 0.83 |
| | 5646–24,404 | 19.42 | 15.71 | 0.81 |
| Distance to stream (m) | 0–997 | 19.41 | 22.76 | 1.17 |
| | 997–2160 | 20.14 | 24.04 | 1.19 |
| | 2160–3573 | 20.98 | 22.12 | 1.05 |
| | 3573–5651 | 19.75 | 15.71 | 0.80 |
| | 5651–21,192 | 19.72 | 15.38 | 0.78 |
| Land use | Open water | 2.19 | 0.08 | 0.04 |
| | Developed, open space | 1.80 | 1.07 | 0.59 |
| | Developed, low intensity | 0.84 | 0.44 | 0.52 |
| | Developed, medium intensity | 0.61 | 0.22 | 0.37 |
| | Developed, high intensity | 0.18 | 0.06 | 0.32 |
| | Deciduous forest | 0.48 | 0.20 | 0.41 |
| | Evergreen forest | 52.09 | 56.72 | 1.09 |
| | Mixed forest | 0.24 | 0.24 | 1.00 |
| | Shrub/scrub | 19.34 | 18.02 | 0.93 |
| | Herbaceous | 9.35 | 22.86 | 2.44 |
| | Cultivated crops | 10.51 | 0.01 | 0.00 |
| | Woody wetlands | 0.50 | 0.07 | 0.14 |
| | Emergent herbaceous wetlands | 1.55 | 0.02 | 0.01 |
| | Barren land | 0.06 | 0.00 | 0.00 |
| | Hay/pasture | 0.26 | 0.00 | 0.00 |

**Table 5.** *Cont.*

| Factor | Class | Total % | Event % | Frequency Ratio |
|---|---|---|---|---|
| Drought index | −1.88−−0.83 | 19.46 | 18.91 | 0.97 |
| | −0.84−−0.32 | 20.38 | 18.59 | 0.91 |
| | −0.33−−0.08 | 19.73 | 9.29 | 0.47 |
| | −0.09−0.21 | 21.15 | 26.60 | 1.26 |
| | 0.21−0.89 | 19.28 | 26.60 | 1.38 |
| NDVI | 0−0.25 | 20.00 | 17.63 | 0.88 |
| | 0.25−0.46 | 20.00 | 16.67 | 0.83 |
| | 0.46−0.61 | 20.01 | 20.83 | 1.04 |
| | 0.61−0.76 | 20.00 | 22.12 | 1.10 |
| | 0.76−1 | 19.98 | 22.76 | 1.14 |
| Plan curvature | Concave | 33.85 | 38.14 | 1.13 |
| | Flat | 16.10 | 11.86 | 0.74 |
| | Convex | 50.05 | 50.00 | 1.00 |
| Precipitation (mm) | 13.93−25.21 | 20.01 | 17.31 | 0.87 |
| | 25.21−33.72 | 20.00 | 24.68 | 1.23 |
| | 33.72−43.35 | 20.00 | 22.76 | 1.14 |
| | 43.35−67.25 | 20.00 | 19.55 | 0.98 |
| | 67.25−116.10 | 19.99 | 15.71 | 0.79 |
| Slope (degree) | 0−1.80 | 19.78 | 12.18 | 0.62 |
| | 1.80−7.22 | 20.59 | 18.59 | 0.90 |
| | 7.22−12.89 | 19.97 | 21.15 | 1.06 |
| | 12.89−20.37 | 19.84 | 19.55 | 0.99 |
| | 20.37−65.75 | 19.82 | 28.53 | 1.44 |
| Soil moisture (mm) | 22.00−95.29 | 19.76 | 21.79 | 1.10 |
| | 95.29−184.29 | 20.16 | 12.82 | 0.64 |
| | 184.29−252.35 | 19.98 | 15.71 | 0.79 |
| | 252.35−348.33 | 19.50 | 25.32 | 1.30 |
| | 348.33−467.00 | 20.60 | 24.36 | 1.18 |
| Solar | 6.14−6.38 | 21.09 | 27.24 | 1.29 |
| | 6.38−6.55 | 21.23 | 27.88 | 1.31 |
| | 6.55−6.72 | 21.00 | 19.55 | 0.93 |
| | 6.72−6.84 | 18.50 | 12.50 | 0.68 |
| | 6.84−7.27 | 18.05 | 12.82 | 0.71 |
| Tmax (°C) | 52.07−59.07 | 20.06 | 12.82 | 0.64 |
| | 59.07−61.84 | 20.01 | 15.71 | 0.78 |
| | 61.84−65.67 | 20.01 | 18.59 | 0.93 |
| | 65.67−74.11 | 19.97 | 30.13 | 1.51 |
| | 74.11−76.06 | 19.95 | 22.76 | 1.14 |
| TWI | 3.29−5.52 | 18.18 | 21.15 | 1.16 |
| | 5.52−6.25 | 20.05 | 22.12 | 1.10 |
| | 6.25−7.27 | 20.88 | 21.47 | 1.03 |
| | 7.27−9.20 | 21.05 | 24.36 | 1.15 |
| | 9.20−26.31 | 19.84 | 10.90 | 0.55 |
| Windspeed (m/s) | 0.99−3.93 | 20.00 | 20.51 | 1.03 |
| | 3.93−4.67 | 20.00 | 21.15 | 1.06 |
| | 4.67−5.29 | 20.00 | 20.19 | 1.00 |
| | 5.29−5.84 | 20.00 | 22.12 | 1.11 |
| | 5.84−11.89 | 20.00 | 16.03 | 0.80 |

### 3.3. Wildfire Susceptibility Map

The construction of the database has been performed, including: (i) the dependent variables (i.e., damage location by wildfire derived by the DPM method and Sentinel-1 product) and (ii) the independent variables (based on topographical, meteorological, environmental, and anthropological). The creation of wildfire susceptibility maps utilizing the training dataset were compiled employing the damage inventory database from the DPM of the wildfires from 2016 to 2020 and applying deep learning with metaheuristic optimization approaches, as discussed above. The wildfire susceptibility maps depict an assessment of the likelihood of experiencing a wildfire in a study area built upon the 16 conditioning factors considered. Figure 7 shows the wildfire susceptibility maps from the CNN, CNN-GWO, and CNN-ICA models. Using the quantile method, we classified

the regions of the wildfire susceptibility maps into five predicted classes: very low, low, moderate, high, and very high [89,90].

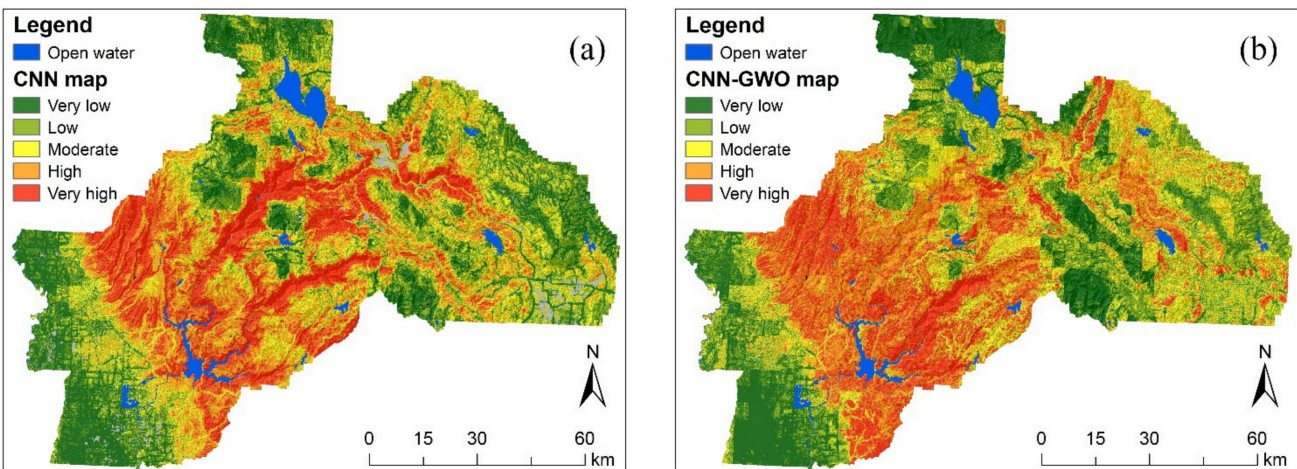

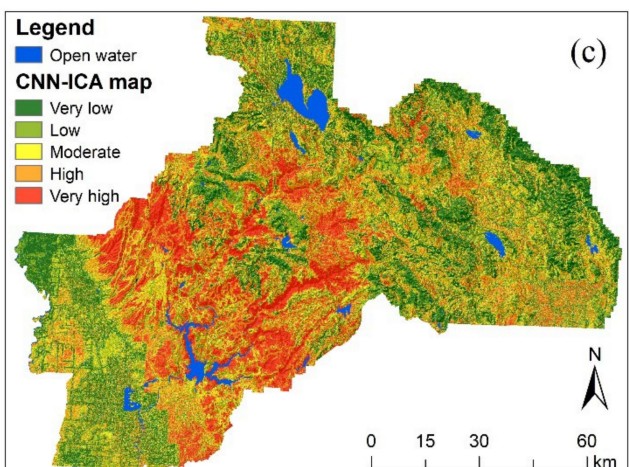

**Figure 7.** Wildfire susceptibility maps generated using 2016–2020 wildfire inventory and the (**a**) CNN, (**b**) CNN-GWO, and (**c**) CNN-ICA algorithms.

The distribution of pixels in each wildfire susceptibility map was also analyzed and is shown in Figure 8. Overall, most of the study area is prone to wildfire occurrences, especially in the western part of Plumas National Forest and expanded to the center of the forest, and makes the study area a wildfire hot-spot region in California. Generally, about 40% of the study area located in areas with flat curvature and low altitude has a low to very low wildfire susceptibility, 20% of the study area has a moderate wildfire susceptibility, and 40% of the study area located in a high degree of slope has a high to very high susceptibility to future wildfire occurrence. In the CNN-ICA model, the moderate class was more distributed throughout the study area than in the other models. The distribution of pixels of standalone CNN and hybrid CNN-GWO showed similar results compared with those of CNN-ICA. CNN-ICA showed the smallest percentage of the very high class and the greatest percentage of the low wildfire susceptibility class.

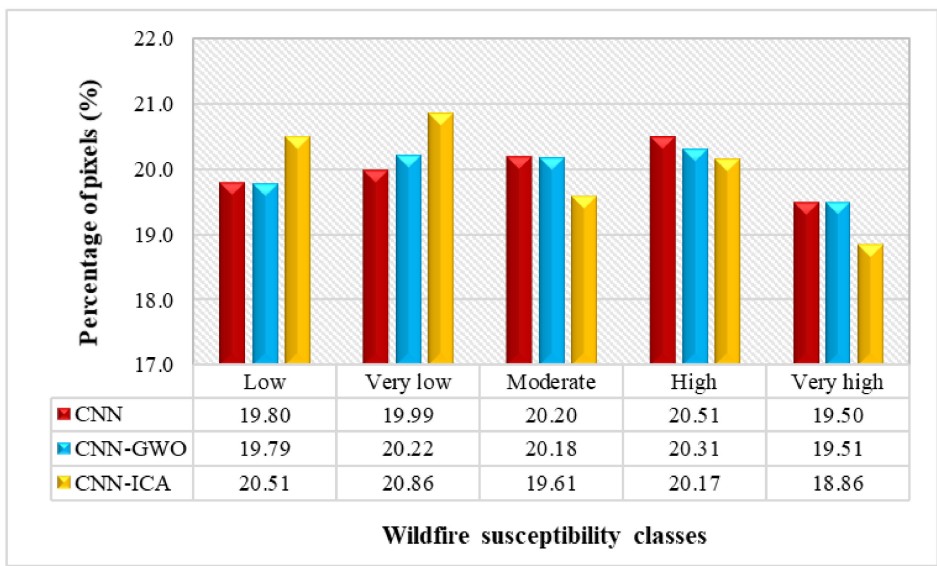

**Figure 8.** The distribution of wildfire susceptibility levels in the CNN, CNN-GWO, and CNN-ICA models.

*3.4. Model Evaluation*

A critical step in generating wildfire susceptibility models is the evaluation of the predicted models to ensure that they provide accurate and reliable wildfire susceptibility assessments. The performance of the models was evaluated and compared to assess the reliability of the wildfire susceptibility maps obtained with each algorithm. Figure 9 has two parts: errors versus number of samples and frequency versus errors. The error part of Figure 9 specifies the values of MSE and RMSE; the frequency versus errors depicts the values of the error mean and standard deviation (StD).

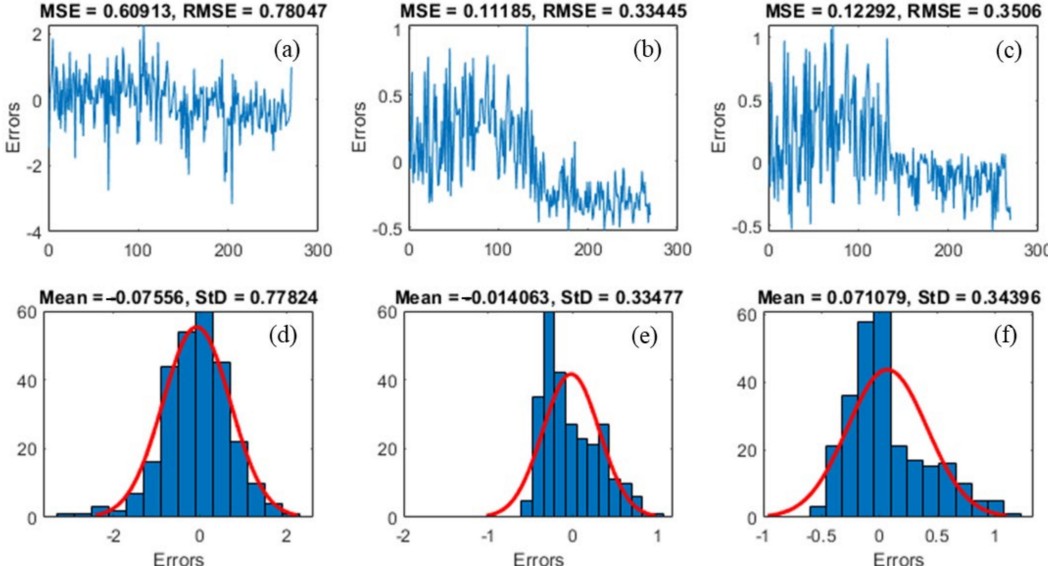

**Figure 9.** Analysis errors based on MSE, RMSE, error mean, and error StD using a testing dataset for: (**a**,**d**) CNN, (**b**,**e**) CNN-GWO, and (**c**,**f**) CNN-ICA models.

The results revealed that in the testing phase using the standalone CNN model, the values of MSE, RMSE, error mean, and error StD are 0.609, 0.780, −0.076, and 0.778, respectively. In the CNN-GWO model, the values of MSE, RMSE, error mean, and error StD are 0.334, 0.112, −0.014, and 0.335, respectively. Moreover, in the CNN-ICA model, the values of MSE, RMSE, error mean, and error StD are 0.129, 0.351, 0.071, and 0.344, respectively. In

addition, the range of error in CNN alone was much broader ($-2 <$ error $< 2$) than that in CNN-GWO and CNN-ICA ($-1 <$ error $< 1$).

Figure 10 shows the ROC curves, with AUC values of 0.934, 0.950, and 0.974 for CNN (red line), CNN-GWO (blue line), and CNN-ICA (green line), respectively. The results of AUC were in agreement with the results of the model evaluation using MSE, RMSE, error mean, and error StD values in the testing phase. Therefore, the predictive ability of CNN-GWO was better than that of CNN and CNN-ICA, as indicated by lower values of MSE and RMSE and a higher value of AUC. This finding is in accordance with the results of other studies [35,91].

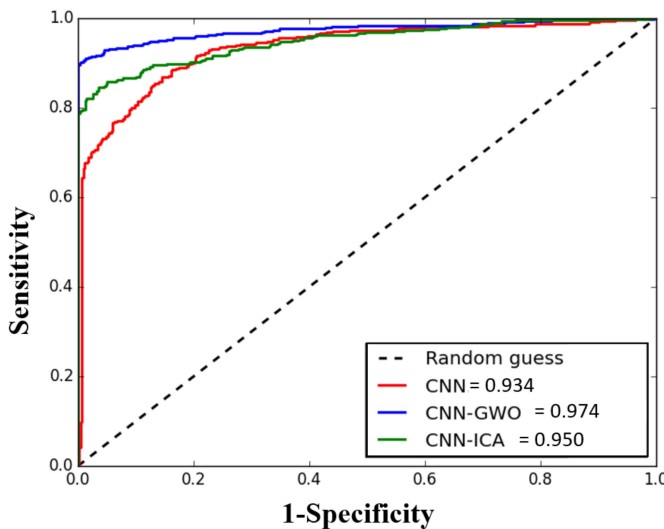

**Figure 10.** The ROC curves and AUC values of CNN (red line), CNN-GWO (blue line), and CNN-ICA (green line) for wildfire susceptibility models using a testing dataset.

## 4. Discussion

Wildfire inventories are crucial for accurate wildfire susceptibility creation. The application of remote sensing SAR imagery is suitable for obtaining data inventories of locations damaged by wildfires due to their wide availability, independence from fieldwork, time and cost efficiency, frequent repeatability over time, cloud and sunlight independence, and high precision and sensitivity [37]. Combined with the fire perimeter generated from CAL FIRE, a more precise and reliable inventory dataset depicting the location of damage and the area of wildfires was obtained. In this study, the DPM technique was employed to determine the areas damaged by wildfires in Plumas National Forest to form inventory data utilized for training and testing the wildfire susceptibility models.

The ARIA team also created a DPM after the Camp Fire wildfire and found similar results [86]. Therefore, using SAR imagery, the DPM method can be an alternative tool to map damaged areas. However, scattered small-colored pixels were spotted outside the perimeter and may be false positives but do not necessarily mean no damage occurred. Our study revealed that the DPM technique is not limited to detecting damage to buildings after an earthquake [57,92], volcano [20], or typhoon [19]. The DPM method can be applied to any natural or anthropogenic damage that causes a substantial change in the surface property as long as a predamage (reference) InSAR pair preserves reasonable coherence. Tay et al. (2020) employed aerial imagery, obtained from the Japan Geospatial Information Authority (GSI), after Typhoon Hagibis. Nur et al. (2021) used a damage area report from the Korea Meteorological Agency (KMA) report of the Gyeongju and Pohang earthquakes.

Important conditioning factors for detecting wildfire-prone areas were identified and collected based on the literature and data availability. Spatial correlation analysis has been performed by calculating the VIF and TOL for multicollinearity analysis and the FR and IGR techniques to evaluate the relationship and influence of the factors. Similar results were acquired by [93], who indicated a higher VIF score for the altitude factor. This is

due to the fluctuation of most factors, such as maximum temperature, drought index, and soil moisture, being consistent with the altitude. The output showed that no significant multicollinearity existed, with all factors used in this study having a considerable effect on wildfire susceptibility in Plumas National Forest. FR results clearly showed that the spatial relationship between each category of conditioning factors and wildfire location is not randomly distributed across Plumas National Forest and that the likelihood of wildfires is highly dependent on the characteristics of the landscapes. Although some variables were not shown to have a significant effect on wildfire occurrence when evaluated separately, as shown in other studies, we were able to identify fire-prone areas in the study area with the help of these conditioning factors. Areas with a high probability of wildfires were associated with land use, drought index, and maximum temperature. These were consistent with many studies that cite temperature, drought index, and precipitation as critical parameters affecting the relative likelihood of wildfire occurrence [14,29], presumably because fuel moisture content is largely a function of temperature, drought index, and precipitation [94]. This study also confirms previous results that found wind is the most unpredictable variable for forecasting wildfire occurrence [24].

Land cover describes the vegetation type, structure, amount, and continuity, with the fuel properties of vegetation and fuel breaks being factors that influence the characteristics of wildfires, such as their likelihood, spread, frequency, and severity, by providing different fuel abundance and settings for different time intervals. In addition, land uses such as urbanization, agriculture, and grazing and land covers such as barren land, open water, and wetland in Plumas National Forest may stop wildfires from spreading because there is insufficient continuous vegetation to carry them. The land use factors in the study area contributed to wildfires, especially in herbaceous areas with the highest FR value and evergreen forest areas with the highest wildfire occurrence. Herbaceous vegetation area is more susceptible to wildfires when the climate is drier, producing fuel, igniting fires, or spreading frequent surface fires [95]. In the forest area, tree leaf litter can also provide fuel when herbaceous groundcover fails to develop a wildfire. The combination of herbaceous vegetation and evergreen forest areas were the primary wildland classes that carried wildfires in the U.S. [96]. In recent decades, the western U.S. has warmed and the frequency and severity of heatwaves and droughts have increased [97]. Based on the National Oceanic and Atmospheric Administration (NOAA), temperatures in California have increased by approximately 3 degrees Fahrenheit (1.67 °C) since the start of the 20th century, with the 2015–2020 period being the warmest, with the highest number of extremely hot days [98]. High temperatures have caused the air to dry out. Fire seasons start earlier and end later each year. Meanwhile, snowpacks are shrinking, causing an earlier spring and an intense and longer dry season. These drier and warmer conditions are also exacerbating wildfires in the western U.S. [9].

In this study, a CNN deep learning algorithm and hybrid models, namely CNN-GWO and CNN-ICA, were used to determine the areas susceptible to wildfires. All proposed models showed good results with AUC values greater than 0.9 and yielded maps of wildfire susceptibility in the study area. The performance of CNN-GWO was the best, with AUC = 0.974 and RMSE = 0.334, followed by CNN-ICA (AUC = 0.950, RMSE = 0.351) and standalone CNN (AUC = 0.934, RMSE = 0.780). In general, the predictive ability of deep learning algorithms can vary depending on the model structure, input selection, dataset quantity and quality, and optimization of model parameters [35]. The hyperparameter adjustment of the CNN model using a metaheuristic algorithm affected the prediction performance of the model. The hybrid models obtained better results than the standalone models. Therefore, hybrid models are important techniques for improving the prediction capacity of basic classification to reduce bias and avoid the problem of overfitting. Furthermore, from the literature reviews, the GWO algorithm has a simple structure and the ability to converge quickly due to the continuous reduction of the search space and fewer tuning variables than other algorithms [77]. Therefore, GWO was both faster and had higher prediction accuracy.

Figure 11 shows the CNN-GWO models which performed better with wildfires that occurred in 2021, including the Beckwourth Complex Fire, Dixie Fire, Gunnison Fire, and Park Fire. The RMSE value for the wildfire susceptibility map from the CNN-GWO model with recent wildfires was 0.511, which was lower than the standalone CNN (0.678) and CNN-ICA (0.514).

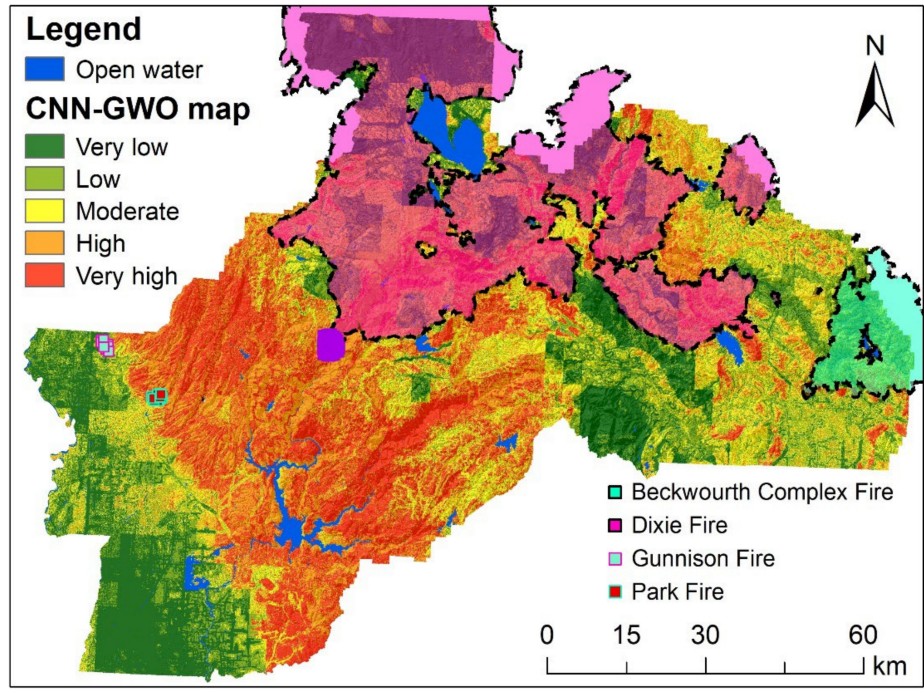

**Figure 11.** Comparison of the CNN-GWO model with wildfires that occurred in 2021.

Gunnison Fire and Park Fire burned 222 ha and 163 ha, respectively. Both wildfires were located in very high wildfire susceptibility areas. The Beckwourth Complex Fire burned 42,700 ha located outside wildfire-prone areas. This is because the model had difficulty predicting the location of wildfires where the area did not meet the required or similar conditions to high-risk areas. Additionally, the area has no history of wildfires. According to the Beckwourth Post-Fire BAER Assessment Report, the cause was lightning. The Dixie Fire burned 389,800 ha, killed one person, and damaged 1329 structures in July 2021. According to the Fire Information for Resource Management System (FIRMS) data, the initial fire was located in an area of very high wildfire susceptibility in the center of the study area (darker purple area in Figure 11). Therefore, our models managed to predict the occurrence of most wildfires in the study area.

This study has the limitation of depicting damaged areas in areas of low coherence before wildfires occurred. Although most wildfires occurred in summer and fall, some forest areas still had dense leaves showing incoherence. Other data should be combined with the DPM and fire perimeter from CAL FIRE to map more precise locations of areas damaged by wildfire. The normalized burn ratio (NBR) from Sentinel-2 imagery with 10 m resolution can reflect the burn severity level. Another limitation is the use of NDVI, solar radiance, drought index, and temperature data. NDVI was computed from MODIS images with 375 m resolution. The NDVI data were influenced by clouds and had certain spatial and temporal limitations. Future study should include the soil adjusted vegetation index (SAVI) and the normalized difference moisture index (NDMI) for more accurate analysis of the vegetation composition in Plumas National Forest [99]. The solar radiance, drought index, and soil moisture data were acquired from Terra Climate images with low resolution. Therefore, the actual climate change in the study area has not been well described, thus introducing uncertainty in the generation of wildfire susceptibility maps. Future study will integrate the capability of Sentinel-1 and Sentinel-2 for better and accurate wildfire mapping.

Along with wildfire susceptibility maps, emergency treatments and activities must be designed to decrease possible impacts on valued resources in the wildfire area, such as life and safety, property, critical natural resources, and cultural resources.

## 5. Conclusions

Wildfire susceptibility maps in Plumas National Forest were created for the first time by integrating SAR data using the DPM method with deep learning based on CNN with metaheuristic optimization algorithms. Identifying areas with very high and high susceptibility provides information that can be applied to risk management and the development of innovative responses to future wildfires. The Sentinel-1 SAR dataset and the DPM technique created a wildfire inventory of the study area with high accuracy and precision as well as low cost and reproducibility. CNN deep learning with metaheuristic optimization algorithms, namely GWO and ICA, were employed to model wildfire susceptibility in Plumas National Forest, California, USA. The results showed that CNN-GWO had a lower RMSE (0.334) and a higher area under the ROC curve value (0.974) than CNN-GWO and CNN alone. In conclusion, the results revealed the applicability of the CNN deep learning algorithm to wildfire susceptibility mapping and the effectiveness of the hybrid approach using metaheuristic optimization algorithms. Furthermore, the methodology is reproducible and applicable to other areas with different conditioning factors. Further study to compare the forecast accuracy of other deep learning methods and metaheuristic optimization algorithms is warranted.

**Author Contributions:** Conceptualization, C.-W.L.; methodology, C.-W.L. and A.S.N.; software, C.-W.L. and A.S.N.; evaluation, C.-W.L., Y.J.K. and A.S.N.; formal analysis, C.-W.L. and A.S.N.; investigation, C.-W.L., Y.J.K. and A.S.N.; resources, C.-W.L., Y.J.K. and A.S.N.; data curation, C.-W.L., Y.J.K. and A.S.N.; writing—original draft preparation, A.S.N.; writing—review and editing, Y.J.K. and C.-W.L.; visualization, C.-W.L. and A.S.N.; supervision, C.-W.L.; project administration, C.-W.L.; funding acquisition, C.-W.L. All authors have read and agreed to the published version of the manuscript.

**Funding:** This research was supported by a grant from the Korea Polar Research Institute (KOPRI, PE22900).

**Data Availability Statement:** Data supporting the reported results can be found at: https://search.asf.alaska.edu/ for Sentinel-1 data, (accessed on 8 July 2022); https://portal.opentopography.org/raster?opentopoID=OTSDEM.032021.4326.3 for Copernicus DEM data, (accessed on 8 July 2022); https://prism.oregonstate.edu/ for precipitation and temperature data, accessed on 8 July 2022; https://www.nrel.gov/gis/solar.html for solar radiance data, accessed on 8 July 2022; https://globalwindatlas.info/ for windspeed data, accessed on 8 July 2022; https://gis.data.ca.gov/ for stream data, accessed on 8 July 2022; https://www.climatologylab.org/terraclimate.html for drought and soil moisture data, accessed on 8 July 2022; https://search.earthdata.nasa.gov/ for NDVI MODIS data, accessed on 8 July 2022; and https://data.usgs.gov/datacatalog/data/USGS:60cb3da7d34e86b938a30cb9 for land use data, accessed on 8 July 2022.

**Conflicts of Interest:** The authors declare no conflict of interest. The funders had no role in the design of the study, in the collection, analyses, or interpretation of data, in the writing of the manuscript, or in the decision to publish the results.

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
