# Peer review of "Creation of Wildfire Susceptibility Maps in Plumas National Forest Using InSAR Coherence, Deep Learning, and Metaheuristic Optimization Approaches"

_remotesensing, doi:10.3390/rs14174416_

Round 1

Reviewer 1 Report

The manuscript “Creation of Wildfire Susceptibility Maps in Plumas National Forest using DPM, Deep Learning, and Metaheuristic Optimization Approaches” is focused on the problem of assessing wildfire susceptibility in California forest, where wildfire is one of the most common natural disasters. Based on Sentinel-1 synthetic aperture radar data collected on images of areas damaged by wildfire (for 2016 to 2020) and applying metaheuristic optimization approaches, a wildfire susceptibility map is generated. The work is based on contemporary mathematical methods and concerns important for target region problem. Regarding the observational methodology, the manuscript is in the scope of the Remote Sensing Journal but it is not constructed in balance, with poor analytical presentation of the results, discussion and conclusion. To be published, major corrections listed below are recommended to be considered.

              Major recommendations:

1.         It seems to me that using abbreviations in the title like “using DPM” is not very relevant; the title should be understandable by a broad community.

2.         The construction of the manuscript is not well balanced: ‘Materials and Methods’ section is too long – 8.5 pages, whereas all other sections (‘Results’, ‘Discussion’, and ‘Conclusion’) are 5 pages in total.

3.         ‘Materials and Methods’ section includes a lot of information, which is not relevant used in the ‘Results’ section, i.e.:

-          Ln 156, Table 1 is missing.

-          Table 1 is first mentioned on Ln 417 in the ‘Results’ section but its content concerns results on collinearity statistics; obviously we have wrong numbering.

-          A special attention is given to wildfire conditioning factors (paragraph 2.3). Figure 4 shows maps of all 16 factors considered to be related to wildfire susceptibility. Why these maps are needed after nowhere in the ‘Results/Discussion’ they are used? For some of the parameters it is not clear on what kind of database these maps are constructed? No information on the nature of drought index – there are many drought indexes, and the same for other parameters?

-          A link/literature source for the origin of each of these parameters to construct maps is needed.

-          Some explanation on each parameter would be useful. In the Results or Conclusion sections short explanation how this contributes the overall susceptibility maps, e.g. in Fig. 6.

-          At the same time, a more detailed explanation of some important methodological aspects would be useful; for example to explain AUC method introduced in paragraph 2.8 as an evaluation metric; how practically AUC method is here applied, please explain.

4.         Results are presented very schematically; analytical description of the reported findings is missing. It would be useful some references on the use of the application of corresponding methods/or construction of fire susceptibility maps in literature to be mentioned if there any. Authors give references for specific applications but do not comment how their results illustrate this literature source, respectively the applications,

-          Ln395, ‘The calculation of FR values in this study are shown in Table 3’; these are not calculations but the results from calculations; you can explain more detailed the procedure of calculation and to comment results more analytically, not only to list the values of parameters as are written in the Table 3.

-          Captions of Tables and Figures should be understandable without reading the text; most of them are not enough informative. Additionally, e.g. for Fig.6 – which period/fire event is used to construct the maps?

-          Because of lot of abbreviations, more explanations with referring to the ‘Materials and Methods’ will help the reading of results; Fig. 8 needs more clear caption.

-          Figure 9, again not clear title; please clarify what does it mean ‘it is threshold-independent’, Ln469? Please comments the meaning of the results not only count the values seen on Tables/Figures.

-          Ln 419 and below, the authors say “no multicollinearities observed between the wildfire conditioning factors”; this is the only a comment, please try to explain how this conclusion has been made and how this is reflected in the results? Also e.g Ln200 “altitude is an important factor affecting the spreading and severity of wildfire; for instance higher moisture levels in highlands prevent …So, soil moisture depends on the altitude and also on the precipitation.

5.         It is highly recommended the existing experience in literature in constructing such maps of fire susceptibility to be presented and some comparison with current results to be proposed.

-          For example the authors may see a recent paper on this topic in Remote Sensing motivating the role of wildfire factors:

Stoyanova JS, Georgiev CG, Neytchev PN. Satellite Observations of Fire Activity in Relation to Biophysical Forcing Effect of Land Surface Temperature in Mediterranean Climate. Remote Sensing. 2022; 14(7):1747. https://doi.org/10.3390/rs14071747 ;

-          Also are there other studies for this region or other regions using this approach or relevant approaches?

6.         Actually, the Discussion section summarizes and repeats again what has been performed; for example,

-          Ln494 ‘Spatial correlation analyses has been done….’, and all factors used in this study had a considerable effect on wildfire susceptibility….’; how would you discuss the spatial distribution of factors?

-          Ln 512, authors cite [9] but no any analyses on susceptibility of herbaceous vegetation and evergreen forest in this study and some other similar points?

-          Please try to discuss your results in the context of the references used.

7.         Conclusion is again a repetition of what has been done and not clear presentation of the main drown conclusions.

8.         It seems that on Ln576 CNN-GWO should be CNN-ICA?

Author Response

The manuscript “Creation of Wildfire Susceptibility Maps in Plumas National Forest using DPM, Deep Learning, and Metaheuristic Optimization Approaches” is focused on the problem of assessing wildfire susceptibility in California forest, where wildfire is one of the most common natural disasters. Based on Sentinel-1 synthetic aperture radar data collected on images of areas damaged by wildfire (for 2016 to 2020) and applying metaheuristic optimization approaches, a wildfire susceptibility map is generated. The work is based on contemporary mathematical methods and concerns important for target region problem. Regarding the observational methodology, the manuscript is in the scope of the Remote Sensing Journal but it is not constructed in balance, with poor analytical presentation of the results, discussion and conclusion. To be published, major corrections listed below are recommended to be considered.

 Response: We appreciate your meticulous review and constructive comments provided by the reviewer and we have made a modification accordingly. Our responses are in red. Line numbers in red refer to line numbers in the revised “Tracked changes” manuscript. Also the changes in manuscript were highlighted in yellow.

              Major recommendations:

  1. It seems to me that using abbreviations in the title like “using DPM” is not very relevant; the title should be understandable by a broad community.

Response: we have changed DPM to InSAR coherence.

Line 2-4, “Creation of Wildfire Susceptibility Maps in Plumas National Forest using InSAR Coherence, Deep Learning, and Metaheu-ristic Optimization Approaches”

  1. The construction of the manuscript is not well balanced: ‘Materials and Methods’ section is too long – 8.5 pages, whereas all other sections (‘Results’, ‘Discussion’, and ‘Conclusion’) are 5 pages in total.

Response: We have tried to extend our analysis in Results and Discussion section to balance the construction of the manuscript.

  1. ‘Materials and Methods’ section includes a lot of information, which is not relevant used in the ‘Results’ section, i.e.:

-          Ln 156, Table 1 is missing.

Response : We are sorry for misunderstanding. Table 1 is located in line 178-179.

-          Table 1 is first mentioned on Ln 417 in the ‘Results’ section but its content concerns results on collinearity statistics; obviously we have wrong numbering.

Response: We have corrected the numbering.

Line 453, “Table 4. IGR and multicollinearity results for the conditioning factors”

-          A special attention is given to wildfire conditioning factors (paragraph 2.3). Figure 4 shows maps of all 16 factors considered to be related to wildfire susceptibility. Why these maps are needed after nowhere in the ‘Results/Discussion’ they are used? For some of the parameters it is not clear on what kind of database these maps are constructed? No information on the nature of drought index – there are many drought indexes, and the same for other parameters?

Response: We have added some explanation in Results about the analysis of all conditioning factors. In discussion section, the conditioning factors were analyzed based on IGR calculation that show the most important factor related to wildfire. The drought index was based on Palmer drought severity index (PDSI), one of the most effective, well-acknowledged, and widely used drought severity index that particularly determines the long-term drought conditions over the forest and other terrestrial ecosystems (Abatzoglou, et al. 2018, Palmer, 1965).

Line 535-537, “The construction of database including: (i) the dependent variables (i.e., damage location by wildfire derived by DPM method and Sentinel-1 product); (ii) the independent variables (based on topographical, meteorological, environmental, and anthropological).”

Line 224-251, “This monthly dataset for global terrestrial surfaces uses the interpolated time-varying anomalies from CRU Ts 4.0/JRA55 with high-spatial climatological normal obtained from WorldClim dataset to create a high-spatial resolution dataset that covers a broader tem-poral record [52]. The drought index based on Palmer Drought Severity Index (PDSI) measures agricultural drought calculated by adding precipitation to the top two layers of the soil and employing a temperature-driven evapotranspiration algorithm to remove moisture [13].”

-          A link/literature source for the origin of each of these parameters to construct maps is needed.

Response: We have added a link or literature source for the origin of each parameter. In data availability statement also provided website link to access the parameter.

Line 215.

Table 3. The list and description of wildfire conditioning factors.

Category

Factors

Scale/

Resolution

Source of data

References

Topography

Aspect

30m

Copernicus DEM

[2];

Altitude

[40];

Slope

[16];

Plan curvature

[14];

Meteorological

Precipitation

800m

PRISM

[2];

Maximum temperature

[39];

Solar

4 km

NREL

[41];

Windspeed

100 m

Global Wind Atlas

[42];

Environmental

Distance to stream

1:5,000

California state geoportal

[16];

Drought index

4 km

Terra climate

[43];

Soil moisture

[44];

NDVI

375 m

MODIS

[39];

Topographic wetness index

30 m

Copernicus DEM

[40];

Anthropological

Land use

30 m

USGS

[2];

Distance to road

[14];

Distance to settlement

[41];

-          Some explanation on each parameter would be useful. In the Results or Conclusion sections short explanation how this contributes the overall susceptibility maps, e.g. in Fig. 6.

Response: He have added some explanation on each parameter in 3.2

Line 447-518, “The slope aspect showed that the wildfire occurrence is concentrated in northeast, southeast, south, southwest, west, and flat portions of Plumas National Forest. The fact that south-facing aspects receive more solar radiation which increase fuel temperature, low fuel moisture in the North Hemisphere and led to wildfire occurrence [50,88]. The ar-eas of 13–272 and 272–1,216 m in altitude had high FR value of about 1.03 and 1.54, re-spectively, indicating that low-altitude areas in the study area are more susceptible to wildfire occurrence and implying vegetation is more burnable due high temperature and dry weather during summer [10,50]. High degree of slope causes the wildfire can spread more quickly up the steep areas, as shown by the two classes of 7.22–12.89 and 20.37–65.75 having FR values 1.06 and 1.44, respectively. Higher distance to road indicates a higher probability of wildfire occurrence, with the three highest classes of 152–301, 301–595, and 595–4,260 having FR values of about 1.19, 1.14, and 1.27, respectively. The fur-ther away the road makes it more difficult for firefighters to access and extinguish the wildfire area [15]. Portions of landscape with maximum temperature > 65.67 oC, higher drought index of -0.09 and precipitation between 25–43.35 mm were predestined places for wildfires, encompassing 52.88% (Tmax > 65.57 oC), 53.20% (drought > -0.09–0.89), and 47.44% (25–43.35 mm precipitation). These results were consistent with many studies that cite temperature, drought index, and precipitation as critical parameters affecting relative likelihood of wildfire occurrence [14,28], presumably because fuel moisture content is largely a function of temperature, drought index, and precipitation [89]. High NDVI clas-ses namely 0.46–0.61, 0.61–0.76, 0.76–1 had FR value of 1.04, 1.10, and 1.14, implying wildfire occurrence in high greenness and moisture content of vegetation. Future study should include soil adjusted vegetation index (SAVI) and normalize difference moisture index (NDMI) for more accurate analysis of the vegetation composition in Plumas Na-tional Forest [90]. The reverse trends were found for distance to settlement with the low classes showing high FR values of 1.28 and 1.30 for distance to settlement 287–1,148 and 1,148–2,679 m. Closer distance to river, namely 0–997, 997–2,160, and 2,160–3,573 had high FR values of about 1.17, 1.19, and 1.05. Rivers are one of the entertaining human in-terests such as tourist camps. The closer distance to settlement and river related to high human activities that increased the risk of wildfire ignition [2,14]. For land use factors, herbaceous, evergreen, and mixed forest classes were experienced highest wildfire occur-rence with the highest FR value which act as fuel for wildfire [10,14]. Plan curvature with concave and convex associated with high occurrence wildfire with FR value, in agreement to other findings that suggest the probability of fire occurrence may be lower on flat terrain and high on concave slopes [91]. The topographical and anthropological features also swamped the effect of soil moisture, TWI, and windspeed, all of which did not show any pattern or specific class that is strongly associated with the wildfire presence or absence. Wildfire occurrence for TWI and windspeed features tend to be high in most classes. Alt-hough our windspeed variable was intended to reflect the outcome of providing fresh ox-ygen and the greater potential for drying potential fuels and propelling fire across land at faster rate. This study confirms previous results that wind is the most unpredictable vari-able for forecasting wildfire occurrence [91].

-          At the same time, a more detailed explanation of some important methodological aspects would be useful; for example to explain AUC method introduced in paragraph 2.8 as an evaluation metric; how practically AUC method is here applied, please explain.

Response: A more explanation has been added on how AUC method applied to evaluate the models.

Line 378-385, “This measure of accuracy range between 0.5 and 1 (perfect forecasting), with values near 1 indicating excellent performance and values near 0.5 denoting very poor prediction accuracy [67]. We applied this method using the testing dataset from wildfire inventory pixels (see Section 3.1) which were not used to train the applied deep learning approaches. A ROC curve is a plot of specificity (i.e., false positive on the x axis) versus sensitivity (i.e., true positive on the y axis). In wildfire modeling, sensitivity and specificity refer to the proportion of correctly predicted fire pixels and the proportion of correctly predicted non-fire pixels, respectively.”

  1. Results are presented very schematically; analytical description of the reported findings is missing. It would be useful some references on the use of the application of corresponding methods/or construction of fire susceptibility maps in literature to be mentioned if there any. Authors give references for specific applications but do not comment how their results illustrate this literature source, respectively the applications,

-          Ln395, ‘The calculation of FR values in this study are shown in Table 3’; these are not calculations but the results from calculations; you can explain more detailed the procedure of calculation and to comment results more analytically, not only to list the values of parameters as are written in the Table 3.

Response: We have rephrased to clarify the sentence. Line 476-477, “The results of FR calculation in this study are shown in Table 5.” The more detailed procedure of calculateion have been explained and the results have been commented analythically (Line 477-526).

Line 468-473, “After classifying the numeric factors into 5 classes, then using tabulate area tools in ArcGIS 10.4 to handle dependent data (wildfire location) with classified factor for obtain-ing tabulate area cell count indicating the number of wildfire occurrence in each class. Therefore, FR value can be acquired by calculating the ratio between the percentage of pix-els wildfire occurrence/event in each class with the percentage of pixels area of each class.”

-          Captions of Tables and Figures should be understandable without reading the text; most of them are not enough informative. Additionally, e.g. for Fig.6 – which period/fire event is used to construct the maps?

Response: We agreed with reviewer, therefore we have changed the caption for better understanding.

Line 548-549, “Figure 7. Wildfire susceptibility maps generated using 2016-2020 wildfire inventory and the (a) CNN, (b) CNN-GWO, and (c) CNN-ICA algorithms.”

-          Because of lot of abbreviations, more explanations with referring to the ‘Materials and Methods’ will help the reading of results; Fig. 8 needs more clear caption.

Response: For better understanding, we have added more explanation for abbreviations such as, MSE=Mean Squared Error, StD= Standard Deviation. Also we have change the caption of Figure 8.

Line 286-387, “This study also used the mean square error (MSE) and root mean squared error (RMSE)…”

Line 575-576. “Figure 9. Analysis errors based on MSE, RMSE, error mean and error StD using testing dataset for (a, d) CNN, (b, e) CNN-GWO, and (c, f) CNN-ICA models.”

-          Figure 9, again not clear title; please clarify what does it mean ‘it is threshold-independent’, Ln469? Please comments the meaning of the results not only count the values seen on Tables/Figures.

Response: We have changed the caption of Figure 9 and we have rephrased to clarify the sentences.

Line 599-600, “Figure 10. The ROC curves and AUC values of CNN (red line), CNN-GWO (blue line), and CNN-ICA (green line) for wildfire susceptibility models using testing dataset.”

-          Ln 419 and below, the authors say “no multicollinearities observed between the wildfire conditioning factors”; this is the only a comment, please try to explain how this conclusion has been made and how this is reflected in the results? Also e.g Ln200 “altitude is an important factor affecting the spreading and severity of wildfire; for instance higher moisture levels in highlands prevent …So, soil moisture depends on the altitude and also on the precipitation.

 Response: The conclusion has been made based on all conditioning factors had VIF scores lower than 10 and TOL scores greater than lower than 0.1. We have rephrased to clarify the the relationship between altitude with moisture level. According to Castro and Chuvieco 1998, altitude influence fuel moisture and air humidity. It has also been reported that fire behavior trends are less severe at higher altitude because of higher rainfall (Chuvieco and Congalton 1988). Therefore, altitude has relationship with moisture level and precipitation.

Line 457-460. “The ranges of VIF and TOL were within the permissible threshold; therefore, there were no multicollinearities observed between the wildfire conditioning factors thereby avoiding the risk of deteriorating model performance.”

  1. It is highly recommended the existing experience in literature in constructing such maps of fire susceptibility to be presented and some comparison with current results to be proposed.

-          For example the authors may see a recent paper on this topic in Remote Sensing motivating the role of wildfire factors:

Stoyanova JS, Georgiev CG, Neytchev PN. Satellite Observations of Fire Activity in Relation to Biophysical Forcing Effect of Land Surface Temperature in Mediterranean Climate. Remote Sensing. 2022; 14(7):1747. https://doi.org/10.3390/rs14071747 ;

Response: Thank you for the reference, we will consider the role of other wildfire factors for future study based on the reference.

-          Also are there other studies for this region or other regions using this approach or relevant approaches?

Response: For the first time, the integration of SAR data using DPM method with deep learning based on CNN with metaheuristic optimization algorithms namely GWO and ICA to create wildfire susceptibility map, especially in Plumas National Forest.

  1. Actually, the Discussion section summarizes and repeats again what has been performed; for example,

-          Ln494 ‘Spatial correlation analyses has been done….’, and all factors used in this study had a considerable effect on wildfire susceptibility….’; how would you discuss the spatial distribution of factors?

Response: We have discussed the spatial distribution of factors based on the highest three IGR values namely landuse, drought index, and maximum temperature with class with highest FR value (Line 629-652).

-          Ln 512, authors cite [9] but no any analyses on susceptibility of herbaceous vegetation and evergreen forest in this study and some other similar points?

Response: We have made analysis on susceptibility of herbaceous vegetation and evergreen forest.

Line 638-644, “The land use factors in the study area contributed to wildfire, especially in herbaceous ar-eas with the highest FR value and evergreen forest areas with the highest wildfire occur-rence. Herbaceous vegetation area is more susceptible to wildfire when the climate is drier to produce fuel, ignite fires, or spread frequent surface fires [103]. In the forest area, tree leaf litter can also provide fuel when herbaceous groundcover fails to develop wildfire. The combination of herbaceous vegetation and evergreen forest areas were the primary wildland classes that carried wildfire in the U.S. [104]”.

-          Please try to discuss your results in the context of the references used.

Response: We have discussed our results in the context of the refences used.

  1. Conclusion is again a repetition of what has been done and not clear presentation of the main drown conclusions.

Response 7: We have rewritten the conclusion.

  1. It seems that on Ln576 CNN-GWO should be CNN-ICA?

Response 8: we would like to clarify that we used CNN-GWO for comparing with the recent wildfire in 2021 due to higher value of AUC and lower value of RMSE.

Reviewer 2 Report

1. I recently saw tons of papers that applied machine learning models to generate wildfire risk maps. Could you emphasize why your study is unique, innovative, and helpful to the community?

2. The wildfire data is usually imbalanced, which is a big issue in building the risk model. Have you considered this factor in this study?

3. The model validation part of this study uses the test portion from the data itself. There are many operational risk maps/fuel maps. I was thinking that compared to those datasets may be more helpful to this study.

4. The conclusion section is more like a summary. Suggest changing the section name to summary.

5. The English need to improve. Many vocabularies and terminology are not appropriate.

Author Response

Reviewer #2.

We would like to appreciate your meticulous review and constructive comments provided by the reviewer and we have made a modification accordingly. Our responses are in red. Line numbers in red refer to line numbers in the revised “Tracked changes” manuscript. Also the changes in manuscript were highlighted in yellow.

  1. I recently saw tons of papers that applied machine learning models to generate wildfire risk maps. Could you emphasize why your study is unique, innovative, and helpful to the community?

Response: Thank you for your comments. The integration of SAR data and fire perimeter map provided accurate data regarding the location of previous wildfires for generating wildfire inventory. SAR satellite has several advantages over optical sensors: clouds and smoke are trans-parent to radar signals; radar does not require sunlight; and due to its coherent character, radar signal has the ability to detect minor alteration in surface property changes due to wildfire. The potential of SAR images for mapping burnt areas lie in the sensitivity of SAR backscattering to vegetation structure and biomass, and the changes in scattering modes caused by fire events. InSAR technique can make a direct measurement on the decorrela-tion among different acquisition dates by integrating both amplitude and phase infor-mation. Furthermore, the use of deep learning algorithm based on convolutional neural network (CNN) with metaheuristic optimization algorithms generated better performance and predictive capability of wildfire susceptibility maps. This is the first time creation of wildfire susceptibility maps in Plumas National Forest using this workflow. Therefore our study is considered novel and is expected helpful to the community.

  1. The wildfire data is usually imbalanced, which is a big issue in building the risk model. Have you considered this factor in this study?

Response: To address the issue, we applied several methods such as Pearson correlation analysis, multicollinearity analysis, information gain ratio (IGR).

  1. The model validation part of this study uses the test portion from the data itself. There are many operational risk maps/fuel maps. I was thinking that compared to those datasets may be more helpful to this study.

Response 3: This is a fair point. Comparison of our result with other operational risk maps/fuels map will help our study. We found that Office of the State Fire Mashal (OSFM) released fire hazard severity zones maps through out California state. However, the latest maps in study area were released in 2007 and 2009. Therefore, it is not relevance.

https://osfm.fire.ca.gov/divisions/community-wildfire-preparedness-and-mitigation/wildland-hazards-building-codes/fire-hazard-severity-zones-maps/

  1. The conclusion section is more like a summary. Suggest changing the section name to summary.

Response 4: As reviewer suggested, we changed the conclusion section to summary.

  1. The English need to improve. Many vocabularies and terminology are not appropriate.

Response 5: Thank you for your meticulous review, the manuscript have been through English correction by expert.

Reviewer 3 Report

To Authors

 The manuscript proposed and produce wildfire susceptibility maps for Plumas National Forest by combining SAR data and deep learning methods based on CNN with metaheuristic optimization algorithms. A wildfire susceptibility map was generated using a CNN and metaheuristic optimization algorithms (GWO and ICA) based on images of areas damaged by wildfires. The locations of damaged areas were identified using the damage proxy map (DPM) technique from Sentinel-1 synthetic aperture radar (SAR) data collected from 2016 to 2020. The wildfire susceptibility models were validated using the area under the receiver operating characteristic (ROC) curve (AUC) and root mean square error (RMSE) analysis. The content is of interest to Satellite Images and aerial Wildfire Susceptibility Map processing for forest fire scientists and remote sensing readers. The manuscript is comprehensive and rich in content. It was well written. I would suggest a minor revision before acceptance. Please check my specific comments as follow for your revisions.   

 Specific comments:

 Firstly, in section 2. Materials and Methods (L 121-129), my question is why don’t authors use water indices derived from remoter sensing data in Environmental factors, it’s very important to study about these indices for example the NDWI can be calculated from remote sensing data and used as an input variable for Environmental factors. Also, why authors did not consider here the MSAVI or SAVI indices for soil properties?

 Secondly, in section 2.1. Study Area (L 134-144), I would like to ask why the authors to add information on the fire season for Plumas National Forest. It’s very important to acknowledge that fire season time for selecting required image dates to eliminate the plant phenology on image analysis and processing. I requested authors clear clarify on that lacking information. Then, (L145-153), while talking about fire events, there is a lack of details time (date, month, year, and times) for each fire events. Authors need to give a table of recent fire events for Plumas National Forest at least from 2016 to 2021 with adding columns on fire areas and forest/non forest cover types for each. Please add my requested information to Figure 3 and add description into paragraphs from current Line 145-153. Figure 3 need to be updated and revised accordingly.

 Thirdly, in Table 1 (L161-163), please cite the data sources for Statistical summary of wildfire ignition causes in Plumas National Forest from 2016 to 2020? How can authors convince the unknown ones?

 Fourth, in Section 2.2 SAR datasets (L164-175) I request authors to add more details on 33 scenes of Sentinel-1 data used. All details of scenes need to be presented here with logic to date of each fire. This is very important to see if the date that satellite images taken are closely to the fire event date or not?

 Fifth, In section 2.3. Wildfire conditioning factors (L177-189), I request the authors to presenting a matrix table showing results for correlations between all variables and explaining why did authors select current parameters or variables in Table 2 (L195).

Sixth, In section 2.4. Damage Proxy Map (DPM) (L242-276), detail for which software authors used to compute DPMs need to be presented.

Finally, in Section 3.4 – Model validation. Why don’t authors use Akaike information criterion (AIC) to assess performance of used models?

Author Response

Reviewer #3

To Authors

 The manuscript proposed and produce wildfire susceptibility maps for Plumas National Forest by combining SAR data and deep learning methods based on CNN with metaheuristic optimization algorithms. A wildfire susceptibility map was generated using a CNN and metaheuristic optimization algorithms (GWO and ICA) based on images of areas damaged by wildfires. The locations of damaged areas were identified using the damage proxy map (DPM) technique from Sentinel-1 synthetic aperture radar (SAR) data collected from 2016 to 2020. The wildfire susceptibility models were validated using the area under the receiver operating characteristic (ROC) curve (AUC) and root mean square error (RMSE) analysis. The content is of interest to Satellite Images and aerial Wildfire Susceptibility Map processing for forest fire scientists and remote sensing readers. The manuscript is comprehensive and rich in content. It was well written. I would suggest a minor revision before acceptance. Please check my specific comments as follow for your revisions.   

Response: We thank the reviewer for this supportive assessment. We have made a modification accordingly Our responses are in red. Line numbers in red refer to line numbers in the revised “Tracked changes” manuscript. Also, the changes in manuscript were highlighted in yellow.

Specific comments:

Firstly, in section 2. Materials and Methods (L 121-129), my question is why don’t authors use water indices derived from remoter sensing data in Environmental factors, it’s very important to study about these indices for example the NDWI can be calculated from remote sensing data and used as an input variable for Environmental factors. Also, why authors did not consider here the MSAVI or SAVI indices for soil properties?

Response: We agreed that water indices such as NDWI are important in wildfire occurrence and development. NDWI can reflect water content in the plant tissue (Lin et al., 2022). Usually remote sensing-derived data were used to provide a quantitative and qualitative approximation of the vegetation cover. Soil Adjusted Vegetation Index (SAVI) that take into account the influence of bare, unsaturated soil backgrounds in order to minimize soil noise. Huete (1988) developed the SAVI (Equation (2)) to minimize soil brightness influences by including a soil adjustment factor L to the NDVI formula so as to correct for soil noise effects (soil variability, color and moisture). Although SAVI accounts for variations in soils, but unlike the NDVI, it is highly sensitive to atmospheric variations and less sensitive to changes in vegetation amount and cover of vegetation greenness (Kalantar et al., 2020). MSAVI is a modified version of SAVI (Soil Adjusted Vegetation Index) developed by Huete, 1988, Qi et al., 1994, respectively. The basic idea of MSAVI is to provide a variable correction factor “L”. This adjustment factor “L” depends on the level of vegetation cover being observed and on the product of NDVI and WDVI (Weighted Difference Vegetation Index). Nevertheless, in this study we consider using NDVI to display the biomass or fuel load through time and across the landscape. Considering the broad spatial and temporal scale of this study, it is more appropriate to use the NDVI variable representing the average biomass condition. In addition, we used soil moisture to display moisture content in the soil. There are many limitations in our study. Further research is needed to use water indices and MSAVI and SAVI for wildfire susceptibility analysis.

Secondly, in section 2.1. Study Area (L 134-144), I would like to ask why the authors to add information on the fire season for Plumas National Forest. It’s very important to acknowledge that fire season time for selecting required image dates to eliminate the plant phenology on image analysis and processing. I requested authors clear clarify on that lacking information. Then, (L145-153), while talking about fire events, there is a lack of details time (date, month, year, and times) for each fire events. Authors need to give a table of recent fire events for Plumas National Forest at least from 2016 to 2021 with adding columns on fire areas and forest/non forest cover types for each. Please add my requested information to Figure 3 and add description into paragraphs from current Line 145-153. Figure 3 need to be updated and revised accordingly.

Response: We have added fire season information based on Figure 3 that has been updated. We also provide detailed wildfire information from 2016 to 2021 such as fire name, fire time, and burned area in Table 2.

Line 163-166. “In general, the peak season for wildfires in Plumas National Forest was late summer and early autumn. In term of frequency, from 2016-2020, the wildfire season started in April, and the most frequent occurrence was observed in August.”

Line 195-196.

Table 2. Wildfire information and Sentinel-1 SAR data used in this study.

Fire name

Alarm date

Burned area (ha)

Pre-event

Post-event

Flight direction

1

2

3

North complex

8/17/2020

129,004

08/01/2020

08/13/2020

09/6/2020

Desc

Sheep

8/17/2020

11,967

08/01/2020

08/13/2020

08/25/2020

Desc

Loyalton

8/14/2020

19,032

08/01/2020

08/13/2020

09/06/2020

Desc

Walker

9/4/2019

22,107

08/20/2019

09/01/2019

09/13/2019

Asc

Camp

11/8/2018

62,053

10/23/2018

11/04/2018

11/16/2018

Desc

Cascade

10/8/2017

4,042

9/10/2017

10/04/2017

10/16/2017

Desc

Cherokee

10/8/2017

3,406

9/10/2017

10/04/2017

10/16/2017

Desc

Ponderosa

8/29/2017

1,625

08/06/2017

08/18/2017

9/11/2017

Asc

Minerva 5

7/29/2017

1,744

07/01/2017

07/13/2017

08/06/2017

Asc

Wall

7/7/2017

2,441

06/18/2017

06/30/2017

07/12/2017

Desc

Saddle

9/5/2016

344

8/5/2016

8/29/2016

9/22/2016

Asc

Thirdly, in Table 1 (L161-163), please cite the data sources for Statistical summary of wildfire ignition causes in Plumas National Forest from 2016 to 2020? How can authors convince the unknown ones?

Response: we have cited the data source and added the description of unknown cause.

Line 167-173, “CAL FIRE [1] also investigated and recorded the cause of the wildfire and found that human causes (direct or indirect) ignited 38.57% of the wildfires in Plumas National For-est (as shown in the statistical summary in Table 1). Therefore, anthropological factors should be considered in constructing a wildfire susceptibility map. According to the re-port, the unknown ignition cause described as a fire that has been investigated and has insufficient information to classify further, a fire that is under investigation, or a fire that has not yet been investigated.”

Fourth, in Section 2.2 SAR datasets (L164-175) I request authors to add more details on 33 scenes of Sentinel-1 data used. All details of scenes need to be presented here with logic to date of each fire. This is very important to see if the date that satellite images taken are closely to the fire event date or not?

Response: We have added a table which provide the information of SAR data used.

Line 195-196,

Table 2. Wildfire information and Sentinel-1 SAR data used in this study.

Fire name

Alarm date

Burned area (ha)

Pre-event

Post-event

Flight direction

1

2

3

North complex

8/17/2020

129,004

08/01/2020

08/13/2020

09/6/2020

Desc

Sheep

8/17/2020

11,967

08/01/2020

08/13/2020

08/25/2020

Desc

Loyalton

8/14/2020

19,032

08/01/2020

08/13/2020

09/06/2020

Desc

Walker

9/4/2019

22,107

08/20/2019

09/01/2019

09/13/2019

Asc

Camp

11/8/2018

62,053

10/23/2018

11/04/2018

11/16/2018

Desc

Cascade

10/8/2017

4,042

9/10/2017

10/04/2017

10/16/2017

Desc

Cherokee

10/8/2017

3,406

9/10/2017

10/04/2017

10/16/2017

Desc

Ponderosa

8/29/2017

1,625

08/06/2017

08/18/2017

9/11/2017

Asc

Minerva 5

7/29/2017

1,744

07/01/2017

07/13/2017

08/06/2017

Asc

Wall

7/7/2017

2,441

06/18/2017

06/30/2017

07/12/2017

Desc

Saddle

9/5/2016

344

8/5/2016

8/29/2016

9/22/2016

Asc

Fifth, In section 2.3. Wildfire conditioning factors (L177-189), I request the authors to presenting a matrix table showing results for correlations between all variables and explaining why did authors select current parameters or variables in Table 2 (L195).

Response 5: As requested, Pearson’s correlation analysis was applied to identify the correlation between all variables.

Line 442-443.

Sixth, In section 2.4. Damage Proxy Map (DPM) (L242-276), detail for which software authors used to compute DPMs need to be presented.

Response 6: We used Gamma software for generating coherence map from SAR data and ArcGIS software for generating DPM.

Finally, in Section 3.4 – Model validation. Why don’t authors use Akaike information criterion (AIC) to assess performance of used models?

Response 7: We consider MSE, RMSE, and area under ROC curve (AUC) analysis were enough to validate the model performance. Further research will consider AIC to assess the performance of used models.

Reviewer 4 Report

Summary

The manuscript presents the development of Wildfire Susceptibility Maps using remote sensing data, conditioning factors and Deep Learning. The study finds better performance of map creation using Deep Learning with Metaheuristic Optimization Approaches. The analysis focuses on California and the data is well used. The introduction is clear and well referenced, although the reasons for using SAR need to be better explained. The method is based on a consistent approached that provides interesting results. The validation is not correctly focused, the authors evaluate the model with the training data, but it is necessary to do a validation with independent data. The results are very interesting but their explanation could be improved. The discussion is not well focused and needs a big improvement. An improved discussion is necessary to offer an adequate study, comparing the analysis with other studies and justifying the results. The conclusions section should be improved. In conclusion, the manuscript presents a relevant study that is very interesting for the scientific community, but the article needs to be greatly improved.

Major comments

Line 77-81. Introduction is clear and well explained but the authors only describe the limitations of optical sensors and the advantages of SAR. Optical has more advantages than SAR in this region, with SAR being more useful in cloudy regions such as tropical areas. In spite of that, the use of SAR without optical is interesting and novel enough to do this study. This manuscript is based on SAR because it is possible to obtain good results with this type of data. Therefore, this study can be very interesting to extrapolate the methodology to other places, and expand the knowledge and possibilities offered by SAR. Please, add the advantages of using optical over SAR and explain the importance of using only SAR. Also, explain what SAR offers that optical does not, it means: What information does SAR provide to the study that optical does not?

Line 165-175. Why use VV and not VV-VH? Or both? Do you use descending or ascending data? Or both? What instrument mode do you use, IW? other parameters? Also, explain the pre-processing and corrections used on the images/data.

Line 173. How much time is there between the images before and after the fire?

Do you think that the different resolutions between data and auxiliary data could interfere with the results?

Line 218. Why do you use mean statistic function and not the median or mode? For example, I recommend the mode for Aspect and Land Cover because the mean biases this type of data, the median for NDVI because it is appropriate for non-parametric data...

Line 272-275. It would be more appropriate in the discussion, not in the results.

Accuracy assessment section. In this section, the authors compare and evaluate the model with the training data. That is not a validation because the authors are not using independent data. Please say that this is a validation is a major error. Carrying out a validation is very important to develop the study and to learn the accuracy of the methodology and maps. Validation is mandatory in a study and this must be done here. In order to carry out a validation, it is necessary to have independent reference data.

Why do you use 50% of the data to train and evaluate? Generally, in machine learning, 2/3 are used to train and 1/3 to evaluate the model.

Line 340-353. The AUC method has several limitations as a metric, and is typically only used in the early steps of model assessments. This metric is not highly recommended for validating or comparing the model. I suggest other metrics like R2 or Mean Absolute Error (MAE).

Line 374-378. It would be more appropriate in the discussion, not in the results.

Line 381-383. It would be more appropriate in the methodology, not in the results.

Line 386. What is zero data?

Line 386-390. Difficult paragraph to understand. Please rewrite.

Table 3. Please explain this table in results. This table offers very interesting results that I suggest to explain in detail. Why are some features higher (Frequency ratio) than others? What does it mean? What does it mean in its factor and globally with the other factors? How important are the higher values within the model? What information do they give to the model?

Line 438-442. Repeat the same information as the figure. That is redundant.

Line 438-442. Explain the results and information.

Line 464. The title of the figures says MSE. Why?

Figure 8. Explain the description of the figure further. What analysis are you showing?

Line 467-473. It would be more adequate in methodology, not in results. In the methods, the authors don’t explain anything about the ROC curve. The ROC curve and the AUC method has several limitations as a metric, and is generally only used in the early steps of model assessments. This metric is not highly recommended for validating or comparing the model. I suggested other metrics.

The results should be further explained and improved, explain the information obtained.

The discussion must be improved, the results must be discussed and compared with other analyses. In this section, the authors describe the results but do not explain them, and the authors do not seek answers as to why this is so. Please, the discussion needs to be greatly improved. The results are very interesting but they must be explained and compared with other studies or data. These findings are not a novelty without adequate explanation. The authors have to explain why based on the results.

In addition, improve the discussion with other literature on susceptibility tests and maps.

Figure 10. It is hard to see the fires.

Figure 10. It is a result and should be explained in the results section (not in discussion). This map and information could be a comparison or a kind of validation of your product which is very interesting.

Figure 10. Could you offer the RMSE values. The map does not seem accurate.

Line 541-550. The remaining fires are in low susceptibility areas. Can you explain that?

Line 555-556. Why not use that information?

Line 556-562. Earlier in the discussion. Extend the paragraph by commenting on the limitations on all variables. And, how do they affect the map and your study?

Line 563-565. Some examples?

Line 575. Better is not a scientific word. I suggest using higher.

Please, the conclusion section needs to be improved a lot. Avoid making a summary.

Line 591-598. Explain or name the different databases used.

Minor comments

Line 11. Include country. Example: ...located in Butte and Plumas counties (USA),...

Figure 2, 4, 5, 6 and 10. It is not necessary to include scale, compass rose, and grid at the same time. Choose between scale and compass rose together, or grid only.

Figure 3, 4. Burned area is usually given in ha or m2, km2.

Figure 4. I suggest sorting by categories. It could be easier and faster to understand.

Figure 4. Captions repeat the title of the figures, which is redundant.

Formula 11. It is formula 9. Please, change it.

Line 408. What is IGR?

Line 417. Change the number. It is Table 4, not 1.

Author Response

Reviewer #4.

Summary

The manuscript presents the development of Wildfire Susceptibility Maps using remote sensing data, conditioning factors and Deep Learning. The study finds better performance of map creation  using Deep Learning with Metaheuristic Optimization Approaches. The analysis focuses on California and the data is well used. The introduction is clear and well referenced, although the reasons for using SAR need to be better explained. The method is based on a consistent approached that provides interesting results. The validation is not correctly focused, the authors evaluate the model with the training data, but it is necessary to do a validation with independent data. The results are very interesting but their explanation could be improved. The discussion is not well focused and needs a big improvement. An improved discussion is necessary to offer an adequate study, comparing the analysis with other studies and justifying the results. The conclusions section should be improved. In conclusion, the manuscript presents a relevant study that is very interesting for the scientific community, but the article needs to be greatly improved.

Response: We appreciate the meticulous review and constructive comments provided by the reviewer and we have made a modification accordingly. Our responses are in red. Line numbers in red refer to line numbers in the revised “Tracked changes” manuscript. Also, the changes in manuscript were highlighted in yellow.

Major comments

Line 77-81. Introduction is clear and well explained but the authors only describe the limitations of optical sensors and the advantages of SAR. Optical has more advantages than SAR in this region, with SAR being more useful in cloudy regions such as tropical areas. In spite of that, the use of SAR without optical is interesting and novel enough to do this study. This manuscript is based on SAR because it is possible to obtain good results with this type of data. Therefore, this study can be very interesting to extrapolate the methodology to other places, and expand the knowledge and possibilities offered by SAR. Please, add the advantages of using optical over SAR and explain the importance of using only SAR. Also, explain what SAR offers that optical does not, it means: What information does SAR provide to the study that optical does not?

Response: Thank you for this supportive assessment, we have added the advantage of optical and SAR. Also, we have added information SAR provide.

Line 81-82, “Sentinel-2 is an optical satellite with 13 spectral bands that can be utilized for fuel condition mapping (e.g., vegetation greenness, fuel moister.) and burn severity mapping.”

Line 83-90, “Radar satellite has several advantages over optical sensors: clouds and smoke are trans-parent to radar signals; radar does not require sunlight; and due to its coherent character, radar signal has the ability to detect minor alteration in surface property changes. The potential of SAR images for mapping burnt areas lie in the sensitivity of SAR backscattering to vegetation structure and biomass, and the changes in scattering modes caused by fire events. InSAR technique can make a direct measurement on the decorrelation among different acquisition dates by integrating both amplitude and phase information [17]”

Line 165-175. Why use VV and not VV-VH? Or both? Do you use descending or ascending data? Or both? What instrument mode do you use, IW? other parameters? Also, explain the pre-processing and corrections used on the images/data.

Response: In this study we used VV polarization and managed to map the burnt areas by achieving lower coherence in new burnt areas and maintain high coherence in unchanged areas. Future study will analyze the capability of VH or combination VV-VH in wildfire mapping. In this study, we used interferometric wide-swath (IW) mode and both ascending and descending data, depends on the availability and shorter temporal baseline with wildfire occurrence. The Sentinel-1 SLC level-1 data were downloaded from Alaska Satellite Facility (ASF) where each scene was pre-processed in Gamma software. The data through orbit information update to reduce orbit error.

Line 187, “We collected Sentinel-1 single look complex (SLC) data with interferometric wide-swath (IW) mode…”

Line 192-194, “The Sentinel-1 SLC level-1 data were downloaded from Alaska Satellite Facility (ASF) where each scene was pre-processed in Gamma software. Each scene was updated through orbit information update to reduce orbit error.”

Line 173. How much time is there between the images before and after the fire?

Response: The time between images before and after the fire is as close as wildfire occurrence date. The date information is provided in Table 2.

Line 195-196, “

Table 2. Wildfire information and Sentinel-1 SAR data used in this study.

Fire name

Alarm date

Burned area (ha)

Pre-event

Post-event

Flight direction

1

2

3

North complex

8/17/2020

129,004

08/01/2020

08/13/2020

09/6/2020

Desc

Sheep

8/17/2020

11,967

08/01/2020

08/13/2020

08/25/2020

Desc

Loyalton

8/14/2020

19,032

08/01/2020

08/13/2020

09/06/2020

Desc

Walker

9/4/2019

22,107

08/20/2019

09/01/2019

09/13/2019

Asc

Camp

11/8/2018

62,053

10/23/2018

11/04/2018

11/16/2018

Desc

Cascade

10/8/2017

4,042

9/10/2017

10/04/2017

10/16/2017

Desc

Cherokee

10/8/2017

3,406

9/10/2017

10/04/2017

10/16/2017

Desc

Ponderosa

8/29/2017

1,625

08/06/2017

08/18/2017

9/11/2017

Asc

Minerva 5

7/29/2017

1,744

07/01/2017

07/13/2017

08/06/2017

Asc

Wall

7/7/2017

2,441

06/18/2017

06/30/2017

07/12/2017

Desc

Saddle

9/5/2016

344

8/5/2016

8/29/2016

9/22/2016

Asc

Do you think that the different resolutions between data and auxiliary data could interfere with the results?

Response: Yes, the different of resolution could interfere the results as seen in Figure 6, precisely in the northern part of the study area. Solar radiance, drought index, and soil moisture data that has spatial resolution of 4 km that not well described the actual climate change in the study area.

Line 690-693, “The solar radiance, drought index, and soil moisture data were acquired from Terra Climate images with low resolution. Therefore, the actual climate change in the study area has not been well described, thus introducing uncertainty in the generation of wildfire susceptibility maps.”

Line 218. Why do you use mean statistic function and not the median or mode? For example, I recommend the mode for Aspect and Land Cover because the mean biases this type of data, the median for NDVI because it is appropriate for non-parametric data...

Response: Agreed, we used mode statistic function for Aspect and Land Use data and median for NDVI data.

To clarify, we have added the explanation in the manuscript.

Line 254-256, “. Instead of using the mean NDVI, we calculated the median NDVI during the 2016 to 2020 on pixel-by-pixel basis to avoid bias introduced to the mean value by the greenness loss after wildfire [54].”

Line 272-275. It would be more appropriate in the discussion, not in the results.

Response: Thank you, we have replaced Line 272-272 in the results to Line 620-623 in the discussion.

Accuracy assessment section. In this section, the authors compare and evaluate the model with the training data. That is not a validation because the authors are not using independent data. Please say that this is a validation is a major error. Carrying out a validation is very important to develop the study and to learn the accuracy of the methodology and maps. Validation is mandatory in a study and this must be done here. In order to carry out a validation, it is necessary to have independent reference data.

Response: We would like to clarify that we used testing data to evaluate the model. Comparison of our result with other operational risk maps/fuels map will help our study. We found that Office of the State Fire Mashal (OSFM) released fire hazard severity zones maps through out California state. However, the latest maps in study area were released in 2007 and 2009. Therefore, it is not relevance.

https://osfm.fire.ca.gov/divisions/community-wildfire-preparedness-and-mitigation/wildland-hazards-building-codes/fire-hazard-severity-zones-maps/

Why do you use 50% of the data to train and evaluate? Generally, in machine learning, 2/3 are used to train and 1/3 to evaluate the model.

Response: We used 50% of the data for training the models and the remaining 50% for evaluation. This consideration based on the abundant number on data of about 71,371 pixels that considered enough for learning phase in CNN deep learning modeling.

Line 340-353. The AUC method has several limitations as a metric, and is typically only used in the early steps of model assessments. This metric is not highly recommended for validating or comparing the model. I suggest other metrics like R2 or Mean Absolute Error (MAE).

Response: We agreed that AUC method has several limitations. Therefore, we used MSE and RMSE analysis along with AUC method for validating the model.

Line 374-378. It would be more appropriate in the discussion, not in the results.

Response: Replaced.

Line 381-383. It would be more appropriate in the methodology, not in the results.

Response: Replaced.

Line 386. What is zero data?

Response: The zero data indicate non-wildfire location and low probability class area that identified using frequency ratio method. These zero data were required for configuring final training and testing data for deep learning modeling and evaluation process.

Line 419-424, “In the context of deep learning, creation of wildfire susceptibility map required data of areas without wildfires. In this study, same number of non-wildfire location data (71,371 points) were sampled through random selection by comparing the area with no prior wildfire and the very low probability class identified with the frequency ratio method. The analysis of zero data based on these results was an efficient way to aid the interpretation of the area [35].”

Line 386-390. Difficult paragraph to understand. Please rewrite.

Response: We have rephrased the paragraph for better understanding.

Line 418-428, “These data then randomly divided into ratio 50% training and 50% testing dataset. In the context of deep learning, creation of wildfire susceptibility map required data of areas without wildfires. In this study, same number of non-wildfire location data (71,371 points) were sampled through random selection by comparing the area with no prior wildfire and the very low probability class identified with the frequency ratio method. The analysis of zero data based on these results was an efficient way to aid the interpretation of the area [35]. The non-wildfire data were also divided into training (50%) and testing (50%) dataset. Then the 50% training datasets for wildfire-occurrence and non-wildfire-occurrence loca-tion were merged to generate wildfire susceptibility maps, and the remaining 50% testing data from both datasets were merged for validation of the performance [60,83,89].”

Table 3. Please explain this table in results. This table offers very interesting results that I suggest to explain in detail. Why are some features higher (Frequency ratio) than others? What does it mean? What does it mean in its factor and globally with the other factors? How important are the higher values within the model? What information do they give to the model?

Response: We have added explanation in detail regarding Table 3.

Line 467-526, “FR values can provide information about the connection between the location of damage caused by wildfire and classes of conditioning factors. After classifying the numeric factors into 5 classes, then using tabulate area tools in ArcGIS 10.4 to handle de-pendent data (wildfire location) with classified factor for obtaining tabulate area cell count indicating the number of wildfire occurrence in each class. Therefore, FR value can be ac-quired by calculating the ratio between the percentage of pixels wildfire occurrence/event in each class with the percentage of pixels area of each class. If a specific class of the factor obtained a FR value greater than 1, then the class has high probability of wildfire occurrence, high degree of damage by wildfire, and will has a substantial impact on wildfire susceptibility [93]. The results of FR calculation in this study are shown in Table 5. The slope aspect showed that the wildfire occurrence is concentrated in northeast, southeast, south, southwest, west, and flat portions of Plumas National Forest. The fact that south-facing aspects receive more solar radiation which increase fuel temperature, low fuel moisture in the North Hemisphere and led to wildfire occurrence [51,94]. The areas of 13–272 and 272–1,216 m in altitude had high FR value of about 1.03 and 1.54, respectively, indicating that low-altitude areas in the study area are more susceptible to wildfire occur-rence and implying vegetation is more burnable due high temperature and dry weather during summer [10,51]. High degree of slope causes the wildfire can spread more quickly up the steep areas, as shown by the two classes of 7.22–12.89 and 20.37–65.75 having FR values 1.06 and 1.44, respectively. Higher distance to road indicates a higher probability of wildfire occurrence, with the three highest classes of 152–301, 301–595, and 595–4,260 having FR values of about 1.19, 1.14, and 1.27, respectively. The further away the road makes it more difficult for firefighters to access and extinguish the wildfire area [15]. Por-tions of landscape with maximum temperature > 65.67 oC, higher drought index of -0.09 and precipitation between 25–43.35 mm were predestined places for wildfires, encom-passing 52.88% (Tmax > 65.57 oC), 53.20% (drought > -0.09–0.89), and 47.44% (25–43.35 mm precipitation). These results were consistent with many studies that cite temperature, drought index, and precipitation as critical parameters affecting relative likelihood of wildfire occurrence [14,29], presumably because fuel moisture content is largely a function of temperature, drought index, and precipitation [95]. High NDVI classes namely 0.46–0.61, 0.61–0.76, 0.76–1 had FR value of 1.04, 1.10, and 1.14, implying wildfire occurrence in high greenness and moisture content of vegetation. Future study should include soil ad-justed vegetation index (SAVI) and normalize difference moisture index (NDMI) for more accurate analysis of the vegetation composition in Plumas National Forest [96]. The re-verse trends were found for distance to settlement with the low classes showing high FR values of 1.28 and 1.30 for distance to settlement 287–1,148 and 1,148–2,679 m. Closer dis-tance to river, namely 0–997, 997–2,160, and 2,160–3,573 had high FR values of about 1.17, 1.19, and 1.05. Rivers are one of the entertaining human interests such as tourist camps. The closer distance to settlement and river related to high human activities that in-creased the risk of wildfire ignition [2,14]. For land use factors, herbaceous, evergreen, and mixed forest classes were experienced highest wildfire occurrence with the highest FR value which act as fuel for wildfire [10,14]. Plan curvature with concave and convex asso-ciated with high occurrence wildfire with FR value, in agreement to other findings that suggest the probability of fire occurrence may be lower on flat terrain and high on concave slopes [97]. The topographical and anthropological features also swamped the effect of soil moisture, TWI, and windspeed, all of which did not show any pattern or specific class that is strongly associated with the wildfire presence or absence. Wildfire occurrence for TWI and windspeed features tend to be high in most classes. Although our windspeed variable was intended to reflect the outcome of providing fresh oxygen and the greater po-tential for drying potential fuels and propelling fire across land at faster rate. This study confirms previous results that wind is the most unpredictable variable for forecasting wildfire occurrence [97].

Our results clearly showed that the spatial relationship between each category of conditioning factors and wildfire location is not randomly distributed across the Plumas National Forest and that the likelihood of wildfire is highly dependent on the characteris-tics of the landscapes. Although some variables were not shown to have significant effect on wildfire occurrence when evaluated separately as shown in other studies, we now able to identify fire prone areas in the study area with the help of these conditioning factors.”

Line 438-442. Repeat the same information as the figure. That is redundant.

Response: Agreed, we have removed the sentences.

Line 438-442. Explain the results and information.

Response: We have added some explanation toward the results.

Line 551-557, “Overall, most of the study area is prone to wildfire occurrences especially in the western part of Plumas National Forest and expanded to the center of the forest and make the study area as wildfire hot-spot region in California. Generally, about 40% of the study area that located in areas with flat curvature and low altitude have a low to very low wildfire susceptibility, 20% of the study area has a moderate wildfire susceptibility, while 40% of the study area located in high degree of slope has a high and very high susceptibility to future wildfire occurrence.”

Line 464. The title of the figures says MSE. Why?

Response: We also calculated MSE value of each model.

Figure 8. Explain the description of the figure further. What analysis are you showing?

Response: We have explained the description and analysis of Figure 9.

Line 570-583, “Figure 9 has two parts including errors versus number of samples and frequency versus errors. The error part of this figure specifies the values of MSE and RMSE. The frequency versus errors depicts the values of error mean and standard deviation (StD). The results revealed that in the testing phase using standalone CNN model, the values of MSE, RMSE, error mean, and error StD are 0.609, 0.780, -0.076, and 0.778, respectively. In the CNN-GWO model, the values of MSE, RMSE, error mean, and error StD are 0.334, 0.112, -0.014, and 0.335, respectively. Moreover, the results expressed that in the CNN-ICA model, the values of MSE, RMSE, error mean, and error StD are 0.129, 0.351, 0.071, and 0.344, respec-tively. In addition, the range of error in CNN alone was much (-2<error<2) broader than that in CNN-GWO and CNN-ICA (-1<error<1).”

Line 467-473. It would be more adequate in methodology, not in results. In the methods, the authors don’t explain anything about the ROC curve. The ROC curve and the AUC method has several limitations as a metric, and is generally only used in the early steps of model assessments. This metric is not highly recommended for validating or comparing the model. I suggested other metrics.

Response: We have replaced the sentences. We agreed that the ROC curve and the AUC method has several limitations as a metric. Therefore, we also calculated the MSE and RMSE for each model.

Line 377-385, “The AUC represents the performance, validation, utilization, and compression of model predictions [84–86]. This measure of accuracy range between 0.5 and 1 (perfect forecast-ing), with values near 1 indicating excellent performance and values near 0.5 denoting very poor prediction accuracy [68]. We applied this method using the testing dataset from wildfire inventory pixels (see Section 3.1) which were not used to train the applied deep learning approaches. A ROC curve is a plot of specificity (i.e., false positive on the x axis) versus sensitivity (i.e., true positive on the y axis). In wildfire modeling, sensitivity and specificity refer to the proportion of correctly predicted fire pixels and the proportion of correctly predicted non-fire pixels, respectively.”

The results should be further explained and improved, explain the information obtained.

Response: We have added some explanation regarding the results.

Line 585-597, “Figure 10 shows that the AUC values were 0.934, 0.950, and 0.974 for CNN (red line), CNN-GWO (blue line), and CNN-ICA (green line), respectively. The results of AUC were in agreement with the results of model validation using MSE, RMSE, error mean, and error StD values in the testing phase. Therefore, the predictive ability of CNN-GWO was better than that of CNN and CNN-ICA, as indicated by a lower value of MSE and RMSE and higher value of AUC. This finding is in accordance with the results of other studies [35,97]. In general, the predictive ability of deep learning algorithms can vary depending on the model structure, input selection, dataset quantity and quality, and optimization of model parameters [98]. The hyper-parameter adjustment of the CNN model using a metaheuristic algorithm affected the prediction performance of the model.”

The discussion must be improved, the results must be discussed and compared with other analyses. In this section, the authors describe the results but do not explain them, and the authors do not seek answers as to why this is so. Please, the discussion needs to be greatly improved. The results are very interesting but they must be explained and compared with other studies or data. These findings are not a novelty without adequate explanation. The authors have to explain why based on the results.

Response: Thank you for your comments, we have made changes in discussion to explain reason based on results and other literatures.

In addition, improve the discussion with other literature on susceptibility tests and maps.

Response: We have discussed results and compared with other literatures.

Figure 10. It is hard to see the fires.

Response: We have made modification to the figure for better visualization.

Figure 11. Comparison of the CNN-GWO model with recent wildfires that occurred in 2021

Figure 10. It is a result and should be explained in the results section (not in discussion). This map and information could be a comparison or a kind of validation of your product which is very interesting.

Response: We consider putting Figure 11 in discussion to discuss the capability and advantage of hybrid deep learning model.

Figure 10. Could you offer the RMSE values. The map does not seem accurate.

Response: We will add RMSE values.

 Line 541-550. The remaining fires are in low susceptibility areas. Can you explain that?

Response: The Beckwourth Complex Fire burned 427 km2 acres located outside wildfire-prone areas. This is because the model had difficulty predicting the location of wildfires where the area has no history of wildfire and did not meet the required or similar conditions to high-risk areas (Line 672-675).

Line 555-556. Why not use that information?

Response: In the future study will integrate the capability of Sentinel-1 and Sentinel-2 for wildfire mapping

Line 556-562. Earlier in the discussion. Extend the paragraph by commenting on the limitations on all variables. And, how do they affect the map and your study?

Response: We have commented the limitation of variables and their effect toward generation wildfire susceptibility map.

Line 682-688, “Another limitation is the use of NDVI, solar radiance, drought index, and temperature da-ta. NDVI was computed from MODIS images with 375 m resolution. The NDVI data were influenced by clouds and had certain spatial and temporal limitations. The solar radiance, drought index, and soil moisture data were acquired from Terra Climate images with low resolution. Therefore, the actual climate change in the study area has not been well de-scribed, thus introducing uncertainty in the generation of wildfire susceptibility maps.”

Line 563-565. Some examples?

Response: The examples such as maximum temperature and drought period in the study area.

Line 575. Better is not a scientific word. I suggest using higher.

Response: Corrected

Please, the conclusion section needs to be improved a lot. Avoid making a summary.

Response: Thank you, we have tried to improve the conclusion.

Line 591-598. Explain or name the different databases used.

Response: We have named the different database used.

Line 728-738, “Data Availability Statement: The data supporting reported results can be found at https://search.asf.alaska.edu/ for Sentinel-1 data, accessed on 8 July 2022,  https://portal.opentopography.org/raster?opentopoID=OTSDEM.032021.4326.3 for Copernicus DEM data, accessed on 8 July 2022, https://prism.oregonstate.edu/ for precipitation and tempera-ture data, accessed on 8 July 2022, https://www.nrel.gov/gis/solar.html for solar radiance data, accessed on 8 July 2022, https://globalwindatlas.info/ for windspeed data, accessed on 8 July 2022, https://gis.data.ca.gov/ for stream data, accessed on 8 July 2022, https://www.climatologylab.org/terraclimate.html for drought and soil moisture data, accessed on 8 July 2022, https://search.earthdata.nasa.gov/ for NDVI MODIS data, accessed on 8 July 2022, https://data.usgs.gov/datacatalog/data/USGS:60cb3da7d34e86b938a30cb9 for land use data, ac-cessed on 8 July 2022.”

Minor comments

Line 11. Include country. Example: ...located in Butte and Plumas counties (USA),...

Response: We have included the country.

Line 146-147, “in Plumas County and the rest extends into Butte County, California, USA.”

Figure 2, 4, 5, 6 and 10. It is not necessary to include scale, compass rose, and grid at the same time. Choose between scale and compass rose together, or grid only.

Response: We have changed the figures by choosing scale and compass rose together.

Figure 3, 4. Burned area is usually given in ha or m2, km2.

Response: We have changed the unit to ha.

Figure 4. I suggest sorting by categories. It could be easier and faster to understand.

Response: As suggested, we have sorted the figures by categories.

Figure 4. Captions repeat the title of the figures, which is redundant.

Response: We have changed the caption.

Line 210-2011, “Figure 4. Wildfire conditioning factors used in this study categorized as (a-d) topographical, (e-h) meteorological, (i-m) environmental, and (n-p) anthropological factors.”

Formula 11. It is formula 9. Please, change it.

Response: Corrected

Line 408. What is IGR?

Response:  IGR stands for information gain ratio, a method to evaluate the wildfire conditioning factors.

Line 307-312, “In this study, information gain ratio (IGR) and multicollinearity analysis were selected to evaluate the wildfire conditioning factors. For the IGR method, the factor with a higher IGR value indicates a stronger prediction ability of the model. However, factors with IGR values equal to or less than 0 indicate a “null” contribution to the forest fire susceptibility model and should be excluded from further analysis [34].”

Line 417. Change the number. It is Table 4, not 1.

Response: Corrected

Line 451, “Table 4. IGR and multicollinearity results for the conditioning factors.”

Round 2

Reviewer 1 Report

Authors have made efforts to improve the manuscript. All recommendations are considered and questions are answered at a satisfactory level.

In this form the manuscript can be published

Author Response

Reviewer 1#

Authors have made efforts to improve the manuscript. All recommendations are considered and questions are answered at a satisfactory level.

In this form the manuscript can be published

Response: Thank you very much for your meticulous review, comments, and positive feedback on the research paper. The paper could improve and express better results.

Reviewer 2 Report

1. So many papers now applied deep learning methods to the wildfire risk field. I don't think it is an innovation. There are several products that have a larger scale that cover the focused area. So selecting Plumas National Forest may not say a novel as well.

2. Climate change is a factor, but I don't think it will cause the operational risk map to be unable to compare due to a different year.

Author Response

Reviewer #2.

  1. So many papers now applied deep learning methods to the wildfire risk field. I don't think it is an innovation. There are several products that have a larger scale that cover the focused area. So selecting Plumas National Forest may not say a novel as well.

Response: We thank the reviewer for the comments and concern related to novelty of our manuscript. Here, we take take the opportunity to clarify the novelty of our study.

1) Plumas National Forest is considered unique since three of the deadliest, largest, and most destructive California wildfires occurred in the study area (Line 50-56). From 2016 to 2021, the burned area was getting wider every year (Line 162-163). Therefore, it is necessary to create wildfire susceptibility map in this specific area.

2) We identify the damage location by wildfire using the advantages and potential of radar satellite (Line 83-91) and DPM method. The results were compared with fire perimeter from CAL FIRE and showed similar damage area after wildfire throughout Plumas National Forest. The combination of both resources resulted a more accurate and dependable wildfire inventory database (Line 430-435).

3) To our knowledge, no such integrated study exist in the literature (Line 700-702). This study provides an intregated approach to map wildfire susceptibility in Plumas National Forest using SAR data and DPM method with deep learning based on CNN with metaheuristic optimization algorithms. The Hybrid model (CNN-GWO and CNN-ICA) showed better performance than standalone CNN based on higher value of AUC and lower value of RMSE (Line 561- 569).

  1. Climate change is a factor, but I don't think it will cause the operational risk map to be unable to compare due to a different year.

Response: Thank you for your comments. This is a fair point that climate change is a long term factor influencing wildfire (Line 635-644). However, temperature, drought index and soil moisture have strong direct and indirect ties to climate variability and climate change. These factors were acquired from PRISM Climate and Terra Climate from 2016 and 2020 (Line 232-234 and Line 243-252). Our results also showed that these factors had relationship with wildfire occurrence, spread, and distribution in the study area based on the IGR (Table 4 and Line 615-620) and FR methods (Table 5 and Line 477-514). Many previous studies also used these factors for wildfire susceptibility mapping [39, 43, 44]. Therefore, these factors are important to be used in this study for creating wildfire susceptibility map.

Reviewer 4 Report

The authors make their efforts to revise the manuscript, and the majority of my comments have been satisfactorily addressed. However, I still have a concern related to the validation. In addition, the results section shows several paragraphs that belong to other sections. The results section should be for displaying results only. No for further explanation of the methodology or to discuss the results with the literature.

Major comments

Line 380-399, 438, 576, 598. Map accuracy analysis is all based on the same database used for training. Despite using different samples for training and testing, it is not an independent database from the one used for training. Therefore, it cannot be called validation. The validation must be done using an independent dataset. If the authors want to do a validation, they must use another database different from the wildfire inventory used or create their own database, mapping the fire perimeters. So I suggest authors call comparison or do proper validation. https://lpvs.gsfc.nasa.gov/

Line 441-449. It would be more appropriate in the methodology, not in the results. The first time the authors mention Pearson correlation analysis is in the results and should be in the methods.

Line 472-474. It would be more appropriate in the discussion, not in the results.

Line 478-486. It would be more appropriate in the methodology, not in the results.

Line 503-507. It would be more appropriate in the discussion, not in the results.

Line 506-511. It would be more appropriate in the discussion, not in the results.

Line 527-529. It would be more appropriate in the discussion, not in the results.

Line 530-536. It would be more appropriate in the discussion, not in the results.

Line 594-595. It would be more appropriate in the methodology, not in the results.

Line 601-605. It would be more appropriate in the discussion, not in the results.

Line 605-607. It would be more appropriate in the methodology, not in the results.

Line 681. Where is the RMSE value of map 10 found?

Minor comments

Figure 2. The remaining figures were changed but this figure was not. It is not necessary to include scale, compass rose, and grid at the same time. Choose between scale and compass rose together, or grid only.

Line163, 165 and 193. Please, coherence between ha and km2, choose one to make it easier to compare between measurements.

Line 388. ROC is defined on line 594, and must be defined on line 388.

Line 458-462 Repeat the same information as the figure. That is redundant.

Line 676. Better is not a scientific word. I suggest using higher.

Author Response

Reviewer #4

The authors make their efforts to revise the manuscript, and the majority of my comments have been satisfactorily addressed. However, I still have a concern related to the validation. In addition, the results section shows several paragraphs that belong to other sections. The results section should be for displaying results only. No for further explanation of the methodology or to discuss the results with the literature.

Response: We appreciate the constructive comments and suggestions from the reviewer. Several improvements have been made based on the reviewer’s comments and suggestions. We hope that our revision can be satisfactory so that our manuscript can be published soon.

Major comments

Line 380-399, 438, 576, 598. Map accuracy analysis is all based on the same database used for training. Despite using different samples for training and testing, it is not an independent database from the one used for training. Therefore, it cannot be called validation. The validation must be done using an independent dataset. If the authors want to do a validation, they must use another database different from the wildfire inventory used or create their own database, mapping the fire perimeters. So I suggest authors call comparison or do proper validation. https://lpvs.gsfc.nasa.gov/

Response: We are grateful for the reviewer’s concern and suggestion about the validation process in this study. Thank you for the website address showing the standardized intercomparison and validation across products from different satellite, algorithms, and agency sources from CEOS Land Product Validation (LPV) mission. To our knowledge, California Department of Forestry and Fire Protection (CAL FIRE) released fire hazard severity zone maps in 2007 (https://osfm.fire.ca.gov/divisions/community-wildfire-preparedness-and-mitigation/wildland-hazards-building-codes/fire-hazard-severity-zones-maps/). Therefore, it is not suitable to compare our results with these data since the year gap is large. Unfortunately, we have no other data to compare our results with so far. We will compare our results with reference in-situ or other suitable reference data based on the standardized validation in the future. Nevertheless, we have changed the term validation into evaluation in the manuscript (Line 126-127, Line 389, Line 446, Line 550). We believe it is more appropriate to use the term evaluation due to the use of same database for training and testing. This evaluation process has been done by many previous studies for creating wildfire susceptibility models, published in high rank journals (Q1).

Zhang, G., Wang, M., Liu, K., Deep neural networks for global wildfire susceptibility modelling, Ecological Indicators, 2021, 127, 107735, https://doi.org/10.1016/j.ecolind.2021.107735.

Piralilou S, Einali G, Ghorbanzadeh O, Nachappa TG, Gholamnia K, Blaschke T, Ghamisi P. A Google Earth Engine Approach for Wildfire Susceptibility Prediction Fusion with Remote Sensing Data of Different Spatial Resolutions. Remote Sensing. 2022; 14(3):672. https://doi.org/10.3390/rs14030672.

Al-Fugara A, Mabdeh AN, Ahmadlou M, Pourghasemi HR, Al-Adamat R, Pradhan B, Al-Shabeeb AR. Wildland Fire Susceptibility Mapping Using Support Vector Regression and Adaptive Neuro-Fuzzy Inference System-Based Whale Optimization Algorithm and Simulated Annealing. ISPRS International Journal of Geo-Information. 2021; 10(6):382. https://doi.org/10.3390/ijgi10060382.

Line 126-127, “Model performance was evaluated using root mean square error (RMSE) analysis…”

Line 389, “Evaluation is an essential step in assessing the accuracy of predictions of a model …”

Line 446, “…testing data from both datasets were merged for evaluation of the performance.”

Line 550, “3.4. Model evaluation”

Line 441-449. It would be more appropriate in the methodology, not in the results. The first time the authors mention Pearson correlation analysis is in the results and should be in the methods.

Response: Thank you for the suggestions. We have replaced to the methods.

Line 309-317, “In addition, Pearson correlation analysis was applied to identify linear correlation relationships between pairs of variables. The correlation coefficient of Pearson allows measurement of the association between variables. When a correlation is present, a change in the magnitude of one variable is associated with a change in the magnitude in the other variable, either in the same direction (positive coefficient) or in the opposite direction (negative coefficient). This coefficient is scaled and takes values between −1 and 1, where 0 is equivalent to the case in which no correlation exists [62]. A correlation coefficient greater than or equal to 0.7 is considered a correlation indicator that can lead to distortion of the modeling process and affect future predictions [63].”

Line 472-474. It would be more appropriate in the discussion, not in the results.

Response: We have replaced to the discussion, as you suggested.

Line 604-607, “The similar results were acquired by [99] who indicated higher VIF score for altitude factor. This is due to the fluctuation of most factors, such as maximum temperature, drought index, and soil moisture are consistent with the altitude.”

Line 478-486. It would be more appropriate in the methodology, not in the results.

Response: We have replaced to the methods.

Line 331-338, “After classifying the numeric factors into 5 classes, then using tabulate area tools in ArcGIS 10.4 to handle dependent data (wildfire location) with classified factor for obtaining tabulate area cell count indicating the number of wildfire occurrence in each class. Therefore, FR value can be acquired by calculating the ratio between the percentage of pixels wildfire occurrence/event in each class with the percentage of pixels area of each class. If a specific class of the factor obtained a FR value greater than 1, then the class has high probability of wildfire occurrence, high degree of damage by wildfire, and will has a substantial impact on wildfire susceptibility [69].”

Line 503-507. It would be more appropriate in the discussion, not in the results.

Response: We have replaced to the discussion.

Line 617-620, “These were consistent with many studies that cite temperature, drought index, and precipitation as critical parameters affecting relative likelihood of wildfire occurrence [14,29], presumably because fuel moisture content is largely a function of temperature, drought index, and precipitation [91].”

Line 506-511. It would be more appropriate in the discussion, not in the results.

Response: We have replaced to the discussion.

Line 687-689, “Future study should include soil adjusted vegetation index (SAVI) and normalize difference moisture index (NDMI) for more accurate analysis of the vegetation composition in Plumas National Forest.”

Line 527-529. It would be more appropriate in the discussion, not in the results.

Response: We have replaced to the discussion.

Line 620-621, “This study also confirms previous results that wind is the most unpredictable variable for forecasting wildfire occurrence [91].”

Line 530-536. It would be more appropriate in the discussion, not in the results.

Response: We have replaced to the discussion.

Line 609-615, “Our results clearly showed that the spatial relationship between each category of conditioning factors and wildfire location is not randomly distributed across the Plumas National Forest and that the likelihood of wildfire is highly dependent on the characteristics of the landscapes. Although some variables were not shown to have significant effect on wildfire occurrence when evaluated separately as shown in other studies, we were able to identify fire prone areas in the study area with the help of these conditioning factors.”

Line 594-595. It would be more appropriate in the methodology, not in the results.

Response: We have replaced to the methods.

Line 391-392, “The ROC curve analysis is a common way of evaluating wildfire probability models.”

Line 601-605. It would be more appropriate in the discussion, not in the results.

Response: We have replaced to the discussion.

Line 650-654, “In general, the predictive ability of deep learning algorithms can vary depending on the model structure, input selection, dataset quantity and quality, and optimization of model parameters [101]. The hyper-parameter adjustment of the CNN model using a metaheuristic algorithm affected the prediction performance of the model.”

Line 605-607. It would be more appropriate in the methodology, not in the results.

Response: We have replaced to the methods.

Line 409-411, “In addition, wildfire susceptibility map based on CNN-GWO will be compared with recent wildfires in 2021 to evaluate the predictive performance of the model.”

Line 681. Where is the RMSE value of map 10 found?

 Response: Thank you for your comments. We have calculated the RSME value between the wildfire susceptibility map with recent wildfires.

Line 663-665, “The RMSE value between the wildfire susceptibility map from CNN-GWO model with re-cent wildfires was 0.511, which was lower than standalone CNN (0.678) and CNN-ICA (0.514).”

Minor comments

Figure 2. The remaining figures were changed but this figure was not. It is not necessary to include scale, compass rose, and grid at the same time. Choose between scale and compass rose together, or grid only.

Response: We appreciate the comment. We have changed the Figure 2 and Figure 5, as you suggest.

Line 152-154, “

Figure 2. The land use of the study area located in California, United States of America, and the distribution of historical wildfires from 2016 to 2020 in Plumas National Forest.”

Line XX, “

Figure 5. DPM depicting damage area after Camp Fire, Walker Fire, North Complex Fire, and Sheep Fire wildfires. Image from Sentinel-2 data acquired on July 17, 2021, during the Dixie Fire.”

Line163, 165 and 193. Please, coherence between ha and km2, choose one to make it easier to compare between measurements.

Response: We have changed the unit with ha.

Line 388. ROC is defined on line 594, and must be defined on line 388.

Response: We have defined the ROC in line 390-392.

Line 390-392, “This study used the area under the receiver operating characteristic (ROC) curve analysis for model assessment, ...”

Line 458-462 Repeat the same information as the figure. That is redundant.

Response: We have rephrased the sentences to avoid redundant.

Line 460-464, “Table 4 shows that the factor with the highest IGR value was land use (IGR=0.39); thus, land use was the most effective wildfire conditioning factor in this study, followed by drought index (0.25) and maximum temperature (0.18). Furthermore, windspeed was found to be the least important wildfire conditioning factor with a value of 0.03.”

Line 676. Better is not a scientific word. I suggest using higher.

Response: We have corrected the sentence as you suggested.

Line 661-663, “Figure 11 shows the CNN-GWO models which performed higher, with recent wildfires that occurred in 2021, including the Beckwourth Complex Fire, Dixie Fire, Gunnison Fire, and Park Fire.”
